# Convex Low-resource Accent-Robust Language Detection in Speech Recognition

**Miria Feng** [1]  **William Tan** [2]  **Mert Pilanci** [1]

## Abstract

Globalization and multiculturalism continue to produce increasingly diverse speech varieties. Yet current spoken dialogue systems frequently fail on under-represented dialects and accents, often misidentifying the input language and causing cascading failures in downstream dialogue tasks. Addressing this dialectal variance under low-resource constraints remains an open challenge, as standard fine-tuning is computationally expensive and prone to overfitting on high-dimensional speech data. We propose Convex Language Detection (CLD), a novel framework that integrates theoretically grounded convex optimization techniques into the spoken dialogue systems pipeline. Our method is efficiently implemented via multi-GPU Alternating Direction Method of Multipliers (ADMM) in JAX, thus providing global optimality guarantees and fast training in polynomial time. Theoretically, we prove that our convex objective induces certified margin stability and provide guarantees against feature perturbations. Empirically, we demonstrate sample efficiency and robustness to input dialectical variation, achieving 97–98% accuracy in challenging low-resource regimes. Our open-source package is available at https://pypi.org/project/jaxcld/.

## 1. Introduction

Spoken language dialogue systems are now ubiquitous across cultures, countries, and applications. From multimodal agents to everyday voice assistants such as Siri (Apple Inc., 2011), Google Assistant (Google LLC, 2016), and Amazon Echo (Amazon.com, Inc., 2014), conversational user interfaces are becoming increasingly essential in daily life. The critical shared component among these systems is Automatic Speech Recognition (ASR), which transcribes user speech input signals into text for downstream processing by Large Language Models (LLMs). Without accurate transcripts, even the most capable LLMs struggle to infer intent or generate reliable responses. This persistent performance gap between speech input and text input has motivated growing interest in developing more robust ASR systems. For example, recent model families such as Whisper (Radford et al., 2023) and Massively Multilingual Speech (MMS; (Pratap et al., 2024)) demonstrate strong zero-shot generalization across domains, yet frequently misidentify input language when confronted with real-world human speech, which presents diverse accents and dialectal variation (Kuhn et al., 2024; Graham & Roll, 2024).

This limitation arises in part since existing voice-transcription datasets rarely annotate fine-grained human speech intonations, leading to systematic under-representation of regional dialects even within high-resource languages. Eberhard et al. (2025) notes approximately 380 million people globally speak English as their first language, with the majority of these speakers using English as a second language, over 600 million people are native Hindi speakers, over 1.3 billion people speak various dialects of Chinese, and over 950 million people speak in various Southeast Asian dialects. One notable example is "Singlish" — the colloquial dialect of Singaporean accented English (Wee, 2018) — whose distinct intonation and prosody (Goh, 2016; Chng, 2003; Rubdy, 2007) results in frequent mistranscription into neighboring languages of Bahasa or Tamil (Le Page, 1984; Rajan, 2018), even by state-of-the-art ASR systems.

Addressing this challenge is technically difficult, since speech patterns vary across age, gender, cultural background, and multilingual experience (Na et al., 2024). Furthermore, spontaneous speech often includes code-switching, disfluencies, and domain-specific vocabulary, all of which further complicate language identification and transcription. These issues create a persistent mismatch between training distributions and real-world applications, which frequently leads to cascading errors in downstream dialogue tasks even as model capacity continues to scale. Given ongoing global trends toward multicultural and multilingual societies, such failures may disproportionately affect millions of users and raise pressing concerns regarding accessibility, inclusion, and user trust (Ngueajio & Washington, 2022; McGuire, 2025). Recent research has begun to

---

[1]Department of Electrical Engineering, [2]Department of Computer Science, Stanford University, California, United States. Correspondence to: Miria Feng <miria00@stanford.edu>.

*Proceedings of the 43rd International Conference on Machine Learning*, Seoul, South Korea. PMLR 306, 2026. Copyright 2026 by the author(s).

examine dialectal gaps across languages (Kantharuban et al., 2023) and explore improved user experiences in multilingual human–computer interaction (Li et al., 2023; Cumbal et al., 2024).

Despite this momentum, progress in spoken dialogue systems continues to lag behind text-only language modeling due to the comparative scarcity of comprehensive voice data. Audio is significantly more expensive to collect and curate than text, requiring strict quality control, privacy considerations, and access to real human participants. As a result, modern ASR performance is increasingly constrained by a lack of available training data (Beaulieu & Leonelli, 2021), and researchers often utilize an unchanging set of established speech corpora (Serban et al., 2015). These resource constraints help explain why, despite rapid gains in LLM scaling capabilities, robust ASR for diverse real-world speech remains a challenging and important open problem. Therefore one current paradigm is to drive progress in speech models by maximizing the signal extracted from limited and low-resource regimes by utilizing sample efficient algorithms (Jimerson et al., 2023).

In this paper, we aim to take a step toward democratizing access to spoken dialogue systems that robustly handle user speech input across multicultural backgrounds. We introduce the **C**onvex **L**anguage **D**etection (**CLD**) framework, which leverages theoretically grounded convex optimization techniques for robust language detection under dialectal variation. Our method achieves global optimality in polynomial time, and demonstrates improved sample efficiency with stronger generalization guarantees. This efficiency is essential for end-to-end spoken dialogue systems, which must maintain sub-500ms latency to preserve natural human conversational timing (Meyer, 2023). We further optimize for fast training and inference by implementing our method in JAX (Bradbury et al., 2021) and solving the foundational convex program using Alternating Direction Method of Multipliers (ADMM) techniques (Boyd et al., 2011). To the best of our knowledge, this represents the first practical application of convex optimization reformulations on speech dialogue systems for language identification.

Our main contributions are summarized as follows:

- We propose **C**onvex **L**anguage **D**etection (CLD), a fast sample-efficient algorithm for robust spoken language classification within low-resource data regimes. We demonstrate CLD's strong efficiency in the critical and challenging dialect-identification task in ASR models. Section 3 formally introduces the CLD algorithm and methodology.

- We recast the CLD network as a convex program and prove certified robustness. By characterizing the variation norm we derive exact logit-Lipschitz constants

and prove certified margin stability against hidden-feature perturbations. This provides a computable, data-dependent certificate of label invariance, ensuring that the model's predictions remain stable within a guaranteed radius in Section 4.

- We validate CLD's empirical performance with expansive experiments across five languages and twenty-four sub-dialects. Notably, CLD remains performant on training datasets with less than one hundred samples. In large model experiments such as Whisper Large v3 and MMS-1B, CLD achieves 97-98% accuracy in low-resource regimes and consistently outperforms competitors. Results are presented in Section 5.

- Our `pip installable` JAX package[1] and open-source code [2] is provided for ease of reproducibility in continued research. This also aims to support global inclusivity efforts and equitable access to speech driven tools by offering a deployable robust plug-in module.

## 2. Related Work

**Multilingual Tasks.** Foundational multilingual ASR models such as Whisper has been trained on more than 99 languages (Radford et al., 2023). However the vast majority of these models perform best on English, with performance dropping significantly on lower resource languages due to lack of training data (Graham & Roll, 2024). This has recently encouraged much work in the field of improving low-resource ASR performance. For example, Bansal et al. (2019), Khare et al. (2021), and Stoian et al. (2020) propose using transfer learning to improve cross-lingual performance. This requires large amounts of speech data in high-resource languages but with text transliterated to the target low-resource language. The mapping serves to encourage increased sharing between the output spaces of both languages, yet the efficacy is not well defined since the languages must share a certain amount of "basis similarity" in linguistics for this to be feasible. During pretraining the base ASR model may also experience catastrophic forgetting, leading to overall deterioration in performance.

**Low-resource Environments.** Even within high resource languages such as English and Mandarin, there exist many dialects which state-of-the-art ASR models struggle to identify correctly. The recent works of Li et al. (2024), Weninger et al. (2019), and Wang et al. (2025) aim to implement prosody-assisted speech systems, or bidirectional Long-Short-Term Memory networks to better model acoustic context. With the rise in popularity of spoken dialogue models, other researchers (Reitmaier et al., 2022) have focused

---

[1] https://pypi.org/project/jaxcld/
[2] https://github.com/pilancilab/CLD

on more clearly identifying the challenges ASR models face with low-resource languages. These methods share the common weakness of being heavily dependent on large fine-tuning datasets with a learning rate that is typically ten times smaller than standard supervised fine-tuning learning rates (Wilson & Martinez, 2001; Liu et al., 2024; de Zuazo et al., 2025).

**Certified Robustness and Lipschitz analysis.** Prior work on robustness certification typically studies the local Lipschitz behavior of already-trained non-convex networks. CLEVER (Weng et al., 2018) formulates adversarial robustness evaluation as local Lipschitz estimation and uses extreme value theory to obtain an attack-agnostic robustness score for large neural networks. Related perturbation-based analyses have also been studied in speech processing, including robustness and privacy settings before downstream alignment (Yang, 2023). Complementary work examines stability under observational interference (Yang et al., 2022) and structural network motifs (Zhang et al., 2025). In contrast, our setting in this work does not estimate a black-box local constant after training. Instead, we leverage the convex reformulation to yield a constructive, data-dependent upper bound on the detection head's variation norm, which directly bounds logit perturbations and gives an explicit hidden-feature margin certificate.

**Convex Programs.** Convex reformulations of two-layer neural networks have been well studied by: Pilanci & Ergen (2020); Bach (2017); Bengio et al. (2005). These reformulations offer polynomial-time convergence to global optima, and seek to mitigate the largely heuristics-driven optimization techniques on non-convex landscapes (Ergen & Pilanci, 2021; Sahiner et al., 2021). However, prior work has largely focused on theoretical properties or small-scale image benchmarks. The recently introduced CRONOS algorithm (Feng et al., 2024) demonstrates promising execution of convex networks for binary language classification tasks at the scale of GPT-2 (Radford et al., 2019). In this work, we scale convex training to multi-class high-dimensional speech representations in data-scarce environments, demonstrating the practical and significant gains of convex methods in real-world spoken dialogue systems.

Related work discussion continues in Appendix B.

# 3. Methodology

The **C**onvex **L**anguage **D**etection (**CLD**) method is formally presented: Section 3.1 provides preliminary background on the convex reformulated program of two-layer ReLU networks, and Section 3.2 presents its integration with ASR model architecture to yield the CLD framework.

## 3.1. Convex Two-Layer ReLU Networks

**Background.** We observe the standard two-layer ReLU network as $f(x) = \sum_{j=1}^{m} (\Theta_{1j} x)_+ \theta_{2j}$, where $j = 1, \ldots, m$ indexes the $m$ hidden units. Here $x \in \mathbb{R}^d$ denotes the input, $\Theta_1 \in \mathbb{R}^{m \times d}$ and $\theta_2 \in \mathbb{R}^m$ are the layer weights, and $(\cdot)_+ = \max\{\cdot, 0\}$ is the ReLU activation. Given training labels $y \in \mathbb{R}^n$, the model's standard non-convex training objective can be seen as

$$\min_{\Theta_1, \theta_2} \ell(f_{\Theta_1, \theta_2}(X), y) + \frac{\beta}{2} \sum_{j=1}^{m} \left( \|\Theta_{1j}\|_2^2 + (\theta_{2j})^2 \right), \quad (1)$$

with loss function $\ell : \mathbb{R}^n \to \mathbb{R}$, data matrix $X \in \mathbb{R}^{n \times d}$, and regularization $\beta \geq 0$. Equation 1 presents a non-convex optimization problem, and its minimization is sensitive to hyperparameter tuning (i.e. learning-rate selection). These issues become amplified in large-scale speech applications, where models are more expensive to train than their text-input counterparts. High-dimensional audio data also practically prohibits comprehensive grid-search of all hyperparameters (Sainath et al., 2013). Our goal is to retain the expressiveness of (1) while employing the stability and reliability of convex methods.

**Convex Reformulation.** Pilanci & Ergen (2020) show that (1) admits an equivalent convex neural network (cvxNN) representation when the hidden width satisfies $m \geq m^*$ for some $m^* \leq n + 1$. The reformulation relies on characterizing all possible ReLU activation patterns induced by $X$. Each pattern corresponds to a diagonal matrix selecting rows of $X$, and the full activation pattern set is $\mathcal{D}_X = \{D = \mathrm{diag}(\mathbf{1}(Xv \geq 0)) : v \in \mathbb{R}^d\}$. The cardinality satisfies $|\mathcal{D}_X| = \mathcal{O}(r(n/r)^r)$ with $r = \mathrm{rank}(X)$ (Pilanci & Ergen, 2020). Given $D_i \in \mathcal{D}_X$, the set of vectors $v$ for which $(Xv)_+ = D_i Xv$, is given by the convex cone:

$$\mathcal{K}_i = \{v \in \mathbb{R}^d : (2D_i - I)Xv \geq 0\}.$$

Exact equivalence between (1) and its convex reformulation requires enumerating all patterns in $\mathcal{D}_X$. In practice, we sample $P$ patterns from $\mathcal{D}_X$ and solve the convex program:

$$\begin{aligned}
\min_{(v_i, w_i)_{i=1}^{P}} \quad & \ell\left( \sum_{i=1}^{P} D_i X(v_i - w_i), y \right) \\
& + \beta \sum_{i=1}^{P} (\|v_i\|_2 + \|w_i\|_2) \\
\text{s.t.} \quad & v_i, w_i \in \mathcal{K}_i, \qquad \forall i \in [P],
\end{aligned} \quad (2)$$

When all patterns are used, (2) retains the same optimal solution as the non-convex formulation of (1) under mild conditions (Mishkin et al., 2022). With sampled patterns,

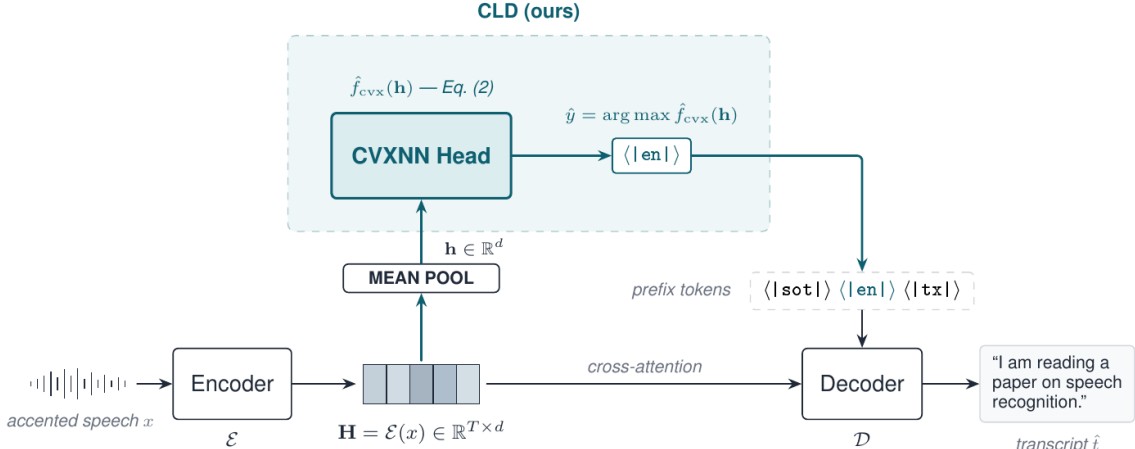

*Figure 1.* CLD Inference (Online)

the solutions may marginally differ. The results of Kim & Pilanci (2024) show that this discrepancy is negligible in practice. CRONOS (Feng et al., 2024) further demonstrated the robustness of convex reformulation strategies on LLM classification tasks (against hyperparameter tuning). Therefore we can confidently leverage this convex surrogate in our study to both preserve the expressive capacity of our models, and enable stable optimization by eliminating dependence on brittle hyperparameters.

### 3.2. Convex Language Detection Algorithm

**Training for Scale.** To tractably work with audio input data our three main algorithmic desiderata are – **scale, speed, and robustness**. Scale is particularly crucial since we require high-dimensional and sensitive multi-class speech data. To address this, we first extract *hidden representations* from the ASR encoder, then solve the resulting problem using a multi-GPU, batched implementation of CRONOS in JAX for enhanced parallelization and load balancing. This yields our CLD framework: a lightweight detection head that operates directly on encoder features to identify the input language before decoding. Formally, given an input waveform $x$ sampled from dataset $(x_i, y_i)_{i=1}^{N}$, the encoder produces a representation $h$ from which CLD predicts a language token $\hat{y}$. The decoder then generates the transcript $\hat{t}$ conditioned on this token. Algorithm 1 outlines the offline CLD training procedure.

**Low Latency Inference.** Algorithm 2 and Figure 1 summarizes the CLD online inference pipeline. Fast online inference is achieved by augmenting the encoder–decoder pipeline (such as Whisper) with the trained convex module. Given an input audio waveform $x$, the encoder $\mathcal{E}$ first produces a series of hidden representations $H$, where we apply masked mean pooling to obtain a fixed-dimensional

---

**Algorithm 1** CLD Training (Offline)

**Input:** Whisper Encoder $\mathcal{E}$, Dataset $\mathcal{D}_{\text{train}}$, parameters $\rho, \beta$
**Output:** Trained Convex Detection Head $\hat{f}_{\text{cvx}}$

1 **for** $i \leftarrow 1$ **to** $N$ **do**
2 $\quad h_i \leftarrow \mathcal{E}(x_i)$
3 Initialize ADMM variables $(\mathbf{v}, \mathbf{w}, \mathbf{u})$.
  **repeat**
4 $\quad (\mathbf{v}, \mathbf{w}) \leftarrow \arg\min_{\mathbf{v}, \mathbf{w}} \Big[ \ell\Big( \sum_{p=1}^{P} D_p X(\mathbf{v}_p - \mathbf{w}_p), y \Big) + \beta \sum_{p=1}^{P} (\|\mathbf{v}_p\|_2 + \|\mathbf{w}_p\|_2) + \frac{\rho}{2}\|v\|_2^2 + \|w\|_2^2 \Big]$

$\quad \mathbf{u} \leftarrow \mathbf{u} + \rho \cdot (\mathbf{v} - \mathbf{w})$
5 **until** *convergence*
6 **return** *Store trained weights as* $\hat{f}_{\text{cvx}}$

---

utterance representation $h$[3], which is passed through the trained convex head $\hat{f}_{\text{cvx}}$ to predict the language token $\hat{y}$. This prediction is computed in a single lightweight forward pass, ensuring sub-500ms latency. The predicted token $\hat{y}$ is then supplied to the Whisper decoder $\mathcal{D}$ as the initialization token, enabling the decoder to accurately generate the final transcription $\hat{t}$ conditioned on the detected language. Thus, CLD integrates seamlessly into the ASR pipeline and improves transcription robustness while incurring negligible latency at inference.

## 4. Theoretical Analysis

This section presents the CLD detection head trained via the convex program in Eq. 2 as Lipschitz-stable in the encoder feature space, and induces a computable robustness

---

[3]This utterance-level design head keeps the module lightweight, easy to integrate into existing ASR pipelines, and performant.

**Algorithm 2** CLD Inference (Online)

**Input:** Whisper Encoder $\mathcal{E}$, Decoder $\mathcal{D}$, Trained Head $\hat{f}_{\mathrm{cvx}}$,
    Input Audio $x$
**Output:** Predicted Language $\hat{y}$, Transcription $\hat{t}$

7   $H \leftarrow \mathcal{E}(x)$
8   $h \leftarrow \mathrm{Pool}(H)$
9   $\hat{y} \leftarrow \arg\max \hat{f}_{\mathrm{cvx}}(h)$
10   $\hat{t} \leftarrow \mathcal{D}(x\,;\,\mathrm{init\_token} = \hat{y})$
11   **return** $(\hat{y}, \hat{t})$

certificate. This validates that bounded perturbations of the encoder output $E(x)$ cause at most linear degradation of the one-vs-rest margin. Therefore any example with sufficiently large initial margin enjoys a certified radius of label invariance. Appendix A provides complete theoretical derivation.

### 4.1. Margin Stability in Hidden Features

The CLD head is formally defined as a multi-class classifier on encoder features, and we quantify how perturbations in those features affect its predictions. Let $E : \mathbb{R}^T \to \mathbb{R}^d$ denote the ASR encoder, and $h = E(x) \in \mathbb{R}^d$ be the hidden features. The CLD detection module $f : \mathbb{R}^d \to \mathbb{R}^K$ is trained by the convex program in Eq. 2. By the cvxNN construction (Section 3.1), the optimal detection head admits a finite two-layer ReLU representation with the same objective value as the convex program up to negligible approximation from activation-pattern sampling. We now introduce the one-vs-rest classification margin and the variation norm, and use them to derive Lipschitz and margin-stability guarantees for $f$.

**Definition 4.1** (One-vs-Rest classification margin)**.** For $y \in \{1, \dots, K\}$ and logits $f(h) = (f_1(h), \dots, f_K(h))$, define the classification margin as

$$\mathrm{mar}(h, y) := f_y(h) - \max_{k \neq y} f_k(h).$$

**Definition 4.2** (Variation norm)**.** The variation norm of $f$ is defined as

$$\|f\|_{\mathrm{var}} := \inf\left\{ \sum_{j=1}^m \|a_j\|_2 \|u_j\|_2 \; : \; f(h) = \sum_{j=1}^m a_j [u_j^\top h]_+ \right\}.$$

**Lemma 4.3** (Logit Lipschitzness)**.** *If $f$ admits a representation of* (2)*, then for any $h, h' \in \mathbb{R}^d$,*

$$\|f(h) - f(h')\|_\infty \leq \|f\|_{\mathrm{var}} \|h - h'\|_2.$$

**Theorem 4.4** (Margin stability under hidden-feature perturbations)**.** *Let $f$ be the detection head given by Eq. 2. For any $y \in \{1, \dots, K\}$ and any $\delta \in \mathbb{R}^d$,*

$$\mathrm{mar}(h + \delta, y) \geq \mathrm{mar}(h, y) - 2\|f\|_{\mathrm{var}} \|\delta\|_2. \quad (3)$$

*Consequently, if $\|\delta\|_2 < \mathrm{mar}(h, y)/(2\|f\|_{\mathrm{var}})$, the predicted class is unchanged.*

Furthermore, if the encoder $E$ is $L_E$-Lipschitz, i.e., $\|E(x) - E(x')\|_2 \leq L_E \|x - x'\|_2$, then

$$\mathrm{mar}(E(x + \eta), y) \geq \mathrm{mar}(E(x), y) - 2\|f\|_{\mathrm{var}} L_E \|\eta\|_2,$$

and the predicted class is preserved whenever $\|\eta\|_2 < \mathrm{mar}(E(x), y)/(2\|f\|_{\mathrm{var}} L_E)$. Proof of Theorem 4.4 is presented in Appendix A.1.

### 4.2. Robustness Certificates from the Convex Program

The previous subsection shows that the variation norm $\|f\|_{\mathrm{var}}$ controls both the logit Lipschitz constant and the stability of the classification margin under hidden-feature perturbations. We now relate this quantity to the solution of Eq. 2. We specifically show that any feasible convex solution $(v_p, w_p)_{p=1}^P$ yields an explicit upper bound on $\|f\|_{\mathrm{var}}$, and that this bound can be expressed in terms of block norms (or Frobenius norms) of the optimization variables. This gives a practical, data-dependent robustness certificate: after training CLD, we can read off a certified Lipschitz constant and margin radius directly from the learned weights.

**Proposition 4.5** (Variation-norm certificate from the convex penalty)**.** *Let $\{(v_p, w_p)\}_{p=1}^P$ denote the variables of Eq. 2. Then*

$$\|f\|_{\mathrm{var}} \leq \mathcal{B}_{\mathrm{cvx}} \quad \text{with} \quad \mathcal{B}_{\mathrm{cvx}} := \sum_{p=1}^P (\|v_p\|_2 + \|w_p\|_2),$$

*interpreting $\|\cdot\|_2$ blockwise (e.g., columnwise $\ell_2$ with a sum across classes). Consequently,*

$$\mathrm{mar}(h + \delta, y) \geq \mathrm{mar}(h, y) - 2\mathcal{B}_{\mathrm{cvx}} \|\delta\|_2.$$

*If the non-convex two-layer form with penalty $\frac{\beta}{2} \sum_j (\|u_j\|_2^2 + \|a_j\|_2^2)$ is used instead, then by AM–GM (Appendix A.5) $\|f\|_{\mathrm{var}} \leq \frac{1}{2} \sum_j (\|u_j\|_2^2 + \|a_j\|_2^2)$, so larger $\beta$ tightens the certified radius.*

In addition, if the logits share blocks (group-sparse outputs), one obtains

$$\|f(h) - f(h')\|_\infty \leq \sum_g w_g \|A_g\|_2 \|U_g\|_2 \|h - h'\|_2,$$

and the margin bound with $\|f\|_{\mathrm{var}}$ can be replaced by the weighted group sum.

**Feature-space certificate.** The certificate above should be interpreted primarily as a hidden-feature certificate. For an encoded utterance $h = E(x)$, define the certified feature-space radius

$$r_h(h, y) = \frac{\mathrm{mar}(h, y)}{2B_{\mathrm{cvx}}}, \quad (4)$$

where $B_{\mathrm{cvx}}$ is the convex penalty-derived upper bound on $\|f\|_{\mathrm{var}}$. Any perturbation $\delta$ satisfying $\|\delta\|_2 < r_h(h, y)$

leaves the predicted language unchanged. If a reliable encoder Lipschitz bound $L_E$ is available, this implies the conservative audio-space radius $r_x = r_h/L_E$. However, for deep Transformer encoders, global $L_E$ bounds can be highly pessimistic, so we report feature-space certificates as the primary stability measure and treat end-to-end audio certificates as conservative diagnostics rather than tight robustness claims.

**Proposition 4.6** (Training-set representation)**.** *Let* $\{(v_i, w_i)\}$ *be any feasible point of* (2)*. Then the training predictions equal those of a (vector-valued) two-layer ReLU network with at most* $2PK$ *hidden units:*

$$f(H) = \sum_{i=1}^{P}\sum_{k=1}^{K}\Big(e_k\,[Hv_{i,k}]_+ \;-\; e_k\,[Hw_{i,k}]_+\Big),$$

*where* $e_k$ *are the standard basis vectors in* $\mathbb{R}^K$*. Equivalently, this network has hidden weights* $\{u_{i,k}^+, u_{i,k}^-\} = \{v_{i,k}, w_{i,k}\}$ *and output weights* $\{a_{i,k}^+, a_{i,k}^-\} = \{e_k, -e_k\}$*.*

**Theorem 4.7** (cvxNN $\Rightarrow$ variation-norm certificate)**.** *Let f be represented as in Proposition 4.6. Then*

$$\|f\|_{\mathrm{var}} \;\le\; \widehat{\mathcal{B}}_{\mathrm{cvx}}^{(2,1)} := \sum_{i=1}^{P}\big(\|v_i\|_{2,1} + \|w_i\|_{2,1}\big).$$

*If* (2) *uses Frobenius penalties instead, then*

$$\|f\|_{\mathrm{var}} \;\le\; \sqrt{K}\,\widehat{\mathcal{B}}_{\mathrm{cvx}}^{\mathrm{F}} \quad with \quad \widehat{\mathcal{B}}_{\mathrm{cvx}}^{\mathrm{F}} := \sum_{i=1}^{P}\big(\|v_i\|_F + \|w_i\|_F\big).$$

Proof of Theorem 4.7 is provided in Appendix A.4.

# 5. Experiments

Section 5.1 provides details on dialect datasets for all experiments. Section 5.2 presents performance evaluation: Word Error Rate (WER) metrics, Character Error Rate (CER), language detection accuracy, wall-clock training time and computational efficiency. Our baseline encoder-decoder ASR models include: Whisper-Small, Whisper-Large-V3, and MMS-1B (Pratap et al., 2024).

We additionally benchmark CLD against the ASR's default language detection as well as a traditional lightweight neural network (NN) model commonly used for language identification in ASR. The NN uses the same encoder embeddings and output labels as CLD, consisting of a linear projection from the encoder dimension to a 256-dimensional hidden layer, followed by a ReLU activation, dropout for regularization, and a final linear layer mapping to the output classes. For the multiclass classification task, we also additionally benchmark against a Support Vector Machine (SVM), Kernel SVM, and $k$-Nearest Neighbors (KNN) Clustering. Experimental details such as hyperparameter grid search are presented in Appendix G.

## 5.1. Speech Datasets

We curate a dataset of multilingual voice transcriptions across high-resource languages and their low-resource sub-dialects. As a primary source of transcription data, we used the Common Voice (v23) dataset (Ardila et al., 2020). We supplement this with several additional dialect datasets for regional speech variance. The Singaporean-English dialect is selected, since studies show it incurs particularly high error rates during voice transcription (Fong et al., 2002). Through the Info-communications and Media Development Authority (Infocomm Media Development Authority, 2025) of Singapore, we were given direct access to the National Speech Corpus (NCS): the first Singapore English corpus. We also use the Lahaja dataset (Javed et al., 2024), a benchmark comprising 12.5 hours of Hindi from 132 speakers across 83 Indian districts. We normalize and augment all audio files via: time stretching, volume gain, pitch shift, and augmented background noise with MUSAN (Snyder et al., 2015). Our experiments span two divisions:

**Binary Classification.** English and Mandarin are the two highest-resource languages in existing speech datasets, which still display some of the lowest accuracy in language prediction for accented speech. This is largely due to the high variance of dialects and accents present (Weninger et al., 2019) in these two languages. For example, Whisper-Small achieves 100% accuracy on Midwestern English, drops to 91.8% accuracy on Wales-accented English, yet only achieves 61.4% accuracy on Malaysian accented English (Table E.6). We select 5 regional dialects per language and perform ablation studies on training sample sizes spanning from 100 to 10 000 samples per language. This quantitatively establishes CLD's performance in low-resource environments. Our experiments equally split training samples across all accents.

**Multiclass Classification.** For the multiclass classification task, we select a total of 5 languages: English, Chinese, Indonesian, Malaysian, Hindi. This selection was made to establish a challenging classification boundary, as these languages share linguistic and geographical proximity. Such regional influences often cause misidentification (e.g. Singaporean English is frequently confused with Malay or Indonesian). To maintain a low-resource experimental setting, we curate a total of 16 000 training samples spanning these 5 languages, thus incorporating 24 unique accents. This resulted in approximately 3200 samples per language and 666 samples per accent. We then conducted a 80-10-10 train, test, validation split.

## 5.2. Main Results and Discussion

**Binary Classification in Low-Resource Regimes.** We validate the performance of CLD augmented Whisper-Small

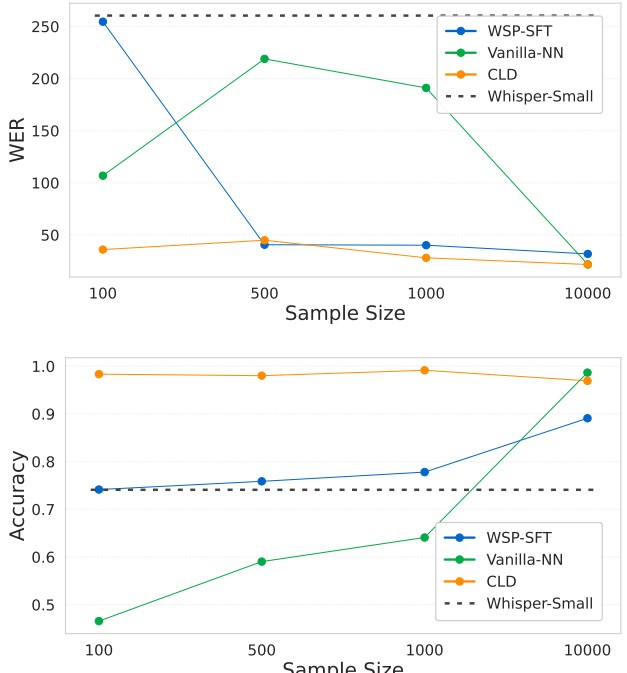

| MODEL | TRAINING TIME (S) | TFLOPs COST |
|---|---|---|
| WSP-SFT | 1,096.74 | 239,528 |
| VANILLA-NN | 840.30 | 183,521 |
| CLD (OURS) | **64.45** | **14,075** |

*Table 1.* Training efficiency of fine-tuned Whisper-Small (WSP-SFT), vanilla-NN, and CLD on the Multiclass dataset. Our method demonstrates substantive compute and training time efficiency.

*Figure 2.* WER (top) — lower is better. Accuracy (bottom) — higher is better. CLD shows robustness and competitive performance regardless of sample size, while competing methods require more data to achieve comparable performance.

in the low-resource setting. Ablation studies span training sample sizes of: 100, 500, 1000, 10 000 [4]. Figure 2 reports seminal WER and language detection accuracies as sample size scales. Both the vanilla NN and the fine-tuned Whisper-Small (WSP-SFT) exhibit a clear correlation where lower WER corresponds with higher detection accuracy as the sample size increases. This validates the intuition that traditional NN require large volumes of data for optimal performance.

Conversely, our CLD model demonstrates visibly consistent performance across all sample sizes, achieving a minimum detection accuracy of 96.94% with 10 000 samples and a maximum of 99.14% with 1000 samples. This validates its high sample efficiency and strong resilience in low-resource settings. CLD also achieves the lowest WER of 21.62 among all models at the 10 000 sample size, thus establishing our method as a promising solution for low-resource regimes. Appendix D provides CER plots.

**Qualitative error analysis.** The large WER reduction should be interpreted primarily as preventing cross-lingual decoding failures rather than fully solving dialectal transcription. In the default Whisper-Small pipeline, accented English utterances can be assigned an incorrect language

token, causing the decoder to produce text in a neighboring or unrelated language. CLD reduces this failure mode by supplying the predicted language token before decoding, allowing the frozen decoder to operate in the correct language space. This explains why WSP WER drops from 139.37 under the default detector to 31.74 with CLD in the multiclass setting, while still leaving room for within-language dialectal transcription errors.

**CLD is Fast and Efficient.** Table 1 shows that the CLD model achieves a training time of just 64.45 seconds—approximately 7.7% of the runtime of a standard vanilla NN, while requiring 13x fewer TFLOPs. This efficiency derives from the convex reformulation solved via ADMM (Appendix C) and implemented in JAX, which enables highly parallelizable updates and rapid convergence. Unlike the vanilla NN which requires multiple passes and steps across the dataset for convergence with necessary hyperparameter grid search, our convex program ensures a unique global optimum with elegant solve methods. Together, these properties establish CLD as both fast and efficient, offering a more practical alternative to conventional largely heuristic-driven performance among traditional architectures.

**Dialect Variation and Performance.** Table 2 summarizes the performance across dialects at the 500 sample size. Additional numerical results as presented in Tables D.1 - D.3 of Appendix D.1. A traditional NN can achieve 100% classification accuracy on a subset of English accents, which is expected given the imbalance of pretraining data and the dominance of English representations in the Whisper feature space. However, this also induces a strong classification bias: the model defaults to predicting English, leading to 88.88% of Chinese samples being misclassified as English. In particular, the NN achieves 8.78% accuracy on Mainland Chinese and 8.84% on Taiwanese dialect, highlighting the English-centric behavior of models trained on imbalanced datasets and the resulting degradation on low-resource dialects.

In contrast, our CLD framework achieves **uniformly high accuracy** across all accents in both languages. For the highly challenging Min Dong Chinese dialect, where the default WSP and fine-tuned NN models suffer significantly

---

[4] We ensure fair comparison and utilize the same 1844 sample size test dataset for evaluation.

| Language - Dialect | Size | Correctly Predicted Samples | | | | Accuracy | | | |
|---|---|---|---|---|---|---|---|---|---|
| | | WSP | WSP-SFT | NN | CLD | WSP | WSP-SFT | NN | CLD |
| EN-Hindi | 190 | 176 | 177 | 190 | 186 | 0.9263 | 0.9316 | **1.0000** | 0.9789 |
| EN-Malaysian | 215 | 136 | 124 | 214 | 214 | 0.6326 | 0.5767 | **0.9953** | **0.9953** |
| EN-Singaporean | 205 | 166 | 162 | 200 | 205 | 0.8098 | 0.7902 | 0.9756 | **1.0000** |
| EN-Pakistani | 189 | 182 | 182 | 189 | 187 | 0.9630 | 0.9630 | **1.0000** | 0.9894 |
| EN-American | 204 | 195 | 194 | 199 | 203 | 0.9559 | 0.9510 | 0.9755 | **0.9951** |
| ZH-Min Dong / Fuzhou | 71 | 7 | 15 | 18 | 63 | 0.0986 | 0.2113 | 0.2535 | **0.8873** |
| ZH-Pu-Xian | 216 | 32 | 56 | 44 | 208 | 0.1481 | 0.2593 | 0.2037 | **0.9630** |
| ZH-Hong Kong | 184 | 121 | 132 | 0 | 174 | 0.6576 | 0.7174 | 0.0000 | **0.9457** |
| ZH-Taiwanese | 181 | 176 | 179 | 16 | 181 | 0.9724 | 0.9890 | 0.0884 | **1.0000** |
| ZH-Mainland | 205 | 187 | 190 | 18 | 202 | 0.9122 | 0.9268 | 0.0878 | **0.9854** |
| Total | 1860 | 1378 | 1411 | 1088 | 1823 | 0.7077 | 0.7207 | 0.5580 | **0.9695** |

*Table 2.* Multi-dialect classification accuracy between English (EN) and Chinese (ZH) across 10 accents for (Whisper-baseline) WSP, WSP-SFT, NN, and CLD at 500 samples per language and 100 samples per dialect.

| Language Classifier | Detection Accuracy | | | WER | | | CER | | |
|---|---|---|---|---|---|---|---|---|---|
| | WSP | WSP-L | MMS-1B | WSP | WSP-L | MMS-1B | WSP | WSP-L | MMS-1B |
| Default | 0.7154 | 0.8033 | 0.6701 | 139.37 | 40.41 | 51.88 | 73.85 | 21.80 | 27.61 |
| KNN | 0.6123 | 0.7145 | 0.4981 | 145.21 | 44.89 | 57.34 | 81.05 | 29.12 | 32.76 |
| Linear SVM | 0.9392 | 0.9501 | 0.5653 | 48.74 | 39.36 | 50.73 | 28.28 | 23.68 | 26.07 |
| Kernel SVM | 0.9431 | 0.9582 | 0.5701 | 46.52 | 37.91 | 49.12 | 26.14 | 22.05 | 25.88 |
| NN | 0.7737 | 0.9605 | 0.8612 | 53.84 | 29.25 | 48.26 | 34.52 | 15.99 | 23.64 |
| CLD (ours) | **0.9715** | **0.9806** | **0.9702** | **31.74** | **28.60** | **45.27** | **17.84** | **15.37** | **21.58** |

*Table 3.* Word Error Rate (WER), Character Error Rate (CER), and Detection Accuracy metrics between Whisper-small (WSP), Whisper-large-v3 (WSP-L), and MMS-1B using KNN, Default, Linear SVM, Kernel SVM, NN, and CLD language classifiers.

(9.86% and 25.35% accuracy, respectively), CLD achieves 88.73%. Across all other dialects, CLD consistently exceeds 94% accuracy. These results demonstrate the sample efficiency, robustness and low variance of CLD in real-world low-resource settings, validating its effectiveness for challenging and diverse dialectal speech.

**Multiclass Classification in Low-Resource Regimes.** Table 3 reports performance on the scaled-up multiclass experiment evaluating additional ASR models—Whisper-Small, Whisper-Large-V3, and MMS-1B—across 5 languages. The standard NN struggles to scale to multiple classes (with the exception of Whisper-Large-V3 at higher training cost), and although the linear SVM and kernel SVM perform well across the Whisper family, they degrade substantially on MMS-1B. We also find that KNN struggles with classification across the board, signaling that simple distance metrics in high-dimensional encoder feature space cannot resolve the challenging dialectal boundaries.

The CLD method achieves the best performance across all evaluation metrics with strong generalization, reaching up to 44.78% increase in detection accuracy and a 12.74% decrease in WER compared to the baseline for MMS-1B. We performed an extensive hyperparameter grid search for all baselines, while CLD is hyperparameter-free due to its

convex formulation, highlighting the practical advantage of guaranteed global optimality over heuristic tuning. For fair comparison we note that larger models like Whisper-Large-V3 and MMS-1B in fact have more accurate language detectors as a baseline. This motivates that augmenting these pretrained models with the modular CLD architecture can increase accuracy and decrease WER by up to 29.21%. These results demonstrate the scaling potential of the CLD model in multiclass settings to improve the performance of larger pretrained models.

Figure 3 presents confusion matrices for our CLD network and vanilla NN on Whisper-Small. The vanilla NN performs well on Hindi (hi) and Indonesian (id) with accuracies of 93% and 98%, yet its performance drops sharply for other languages, reaching only 44% on Chinese (zh). It also exhibits systematic confusions: misclassifying Chinese (zh) as Indonesian (id) in 34% of cases and Malay (ms) as Indonesian (id) in 23%. In contrast, our novel CLD model shows strong diagonal dominance across all languages, with its **lowest accuracy still at 95%** in en, counteracting the English-centric bias. Further discussion on bias among low-resource dialects in high-resource languages continues in Section D.3.

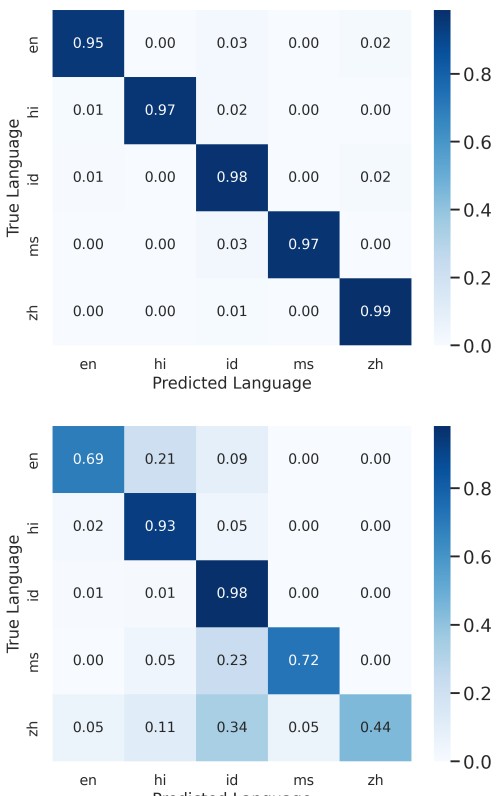

**Concierge:** Hello Mr. Kevin Fong, this is Lucy at the front desk. How may I help you?
**Guest:** Baru keadaan seperti seorang seorang seorang seperti seorang, seorang seorang berada di dalamnya.

*Figure 3.* Language classification performance using Whisper-small across English (en), Chinese (zh), Indonesian (id), Malaysian (ms), and Hindi (hi). CLD (top) is compared against a vanilla neural network (bottom).

**Qualitative Human Case Study.** Our human-facing case study illustrates the user-visible impact of language misidentification in realistic dialogue settings. This study is not intended as a statistically powered population-level evaluation; our primary results are the large-scale benchmarks and analyses. In the case study, participants used a constrained hospitality scenario to ensure the conversational domain remained consistent across systems. The results highlight a recurring qualitative failure mode: default language detection can route accented English or Mandarin utterances into the wrong decoding language, producing cross-lingual transcripts that are unusable.

CLD reduced the number of wrong-language outputs and word errors in this illustrative setting, supporting the quantitative trends reported in the main benchmark. The surveys were conducted with five participants in Singapore speaking English (EN) and ten participants in southeastern China speaking Mandarin (ZH). This experiment underscores the fundamental role of metrics as proxies for qualitative real world studies. Participants were instructed to assume the position of a general guest in a hospitality setting requesting an item to ensure precise and consistent dialogue across all models. One example of Whisper's output is below:

Although the Singaporean participant spoke naturally in his native English, the Whisper model detected and transcribed the input incorrectly into Bahasa. Interestingly, Singaporean English also frequently mis-transcribed Mandarin characters (and vice versa), and we discovered an unexpected type of error also arose from traditional NN detection heads: local accents introduced errors such as the user speaking *'Both hot and cold settings'* to *'Both hood and coat setting'*. In contrast, our CLD framework achieved the fastest inference and most accurate results demonstrating robustness to input dialect variance. CLD achieved both minimal word errors and the smallest number of incorrect language detections. Additional numerical tables of results and performance plots are presented in the Appendix (Table F.7).

## 6. Conclusion

We introduced the Convex Language Detection (CLD) framework as a fast, lightweight, and theoretically grounded method for robust language identification in ASR tasks. Leveraging the convex reformulation yields certified robustness and improved sample efficiency, allowing CLD to maintain consistent, high accuracy regardless of dataset size. CLD effectively mitigated the English-centric failure modes of existing ASR pipelines, demonstrating its effectiveness in addressing low-resource yet diverse dialectal speech conditions. Future work will explore integrating CLD more deeply within the traditional ASR model to enable end-to-end continuous representation learning, and investigate applying this convex-analytic framework within multimodal agentic models.

**End-to-end differentiable CLD.** A natural extension in future work is to make the full pipeline differentiable by back-propagating through the convex program. This can be implemented by unrolling the ADMM iterations or by implicit differentiation through the KKT optimality conditions of Equation (2). Differentiable convex optimization layers (Agrawal et al., 2019) have previously shown that convex programs can be embedded into neural architectures and differentiated end-to-end, suggesting a path toward training encoders whose latent representations are optimized for convex language separability.

**Acknowledgments**

This work was supported in part by the National Science Foundation (NSF) CAREER Award under Grant CCF-2236829, in part by the National Institutes of Health under Grant 1R01AG08950901A1, in part by the Office of Naval Research under Grant N00014-24-1-2164, and in part by the Defense Advanced Research Projects Agency under Grant HR00112490441. In addition, Miria Feng was supported in part by the Stanford Graduate Fellowship.

We thank Lucy Woof, Andrew Maas, and students from the CS224S class at Stanford University for valuable feedback and many insightful discussions.

## Impact statement

Our work introduces a practical, efficient, and effective method for improving accessibility to spoken dialogue models for global users. Spoken dialogue models are becoming increasingly ubiquitous in our daily lives, yet access is often suboptimal due to lack of understanding of input dialect. We aim to take a step towards democratizing models, and increase equitable access to a broader society of diverse speech variations, while encouraging more research in developing machine learning tools for the global audience.

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

# A. Proof of Main Results and certificates for margin stability

This section links the convex program in Eq. 2 and the two-layer ReLU representation together with computable certificates that translate directly into certified radii.

## A.1. Proof of Lemma 4.3

*Proof.* Write $f_k(h) = \sum_j a_{j,k} [u_j^\top h]_+$. Since $t \mapsto [t]_+$ is 1-Lipschitz, $|f_k(h) - f_k(h')| \leq \sum_j |a_{j,k}| |u_j^\top (h - h')| \leq \sum_j \|a_j\|_2 \|u_j\|_2 \|h - h'\|_2$. Maximizing over $k$ and taking the infimum over representations yields the result. $\square$

## Proof of Theorem 4.4

*Proof.* Let $k^\star = \arg\max_{k \neq y} f_k(h)$. Then

$$\mathrm{mar}(h + \delta, y) - \mathrm{mar}(h, y) = \big(f_y(h + \delta) - f_y(h)\big) - \big(\max_{k \neq y} f_k(h + \delta) - f_{k^\star}(h)\big).$$

Each difference is $\leq \|f\|_{\mathrm{var}} \|\delta\|_2$ by Lemma 4.3, yielding (3). $\square$

## A.2. Activation patterns and pattern cones

For $u \in \mathbb{R}^d$ and a data matrix $X \in \mathbb{R}^{n \times d}$, define the training activation pattern

$$D(X, u) = \mathrm{diag}\big(\mathbf{1}\{Xu \geq 0\}\big) \in \{0, 1\}^{n \times n}.$$

Given a fixed pattern $D \in \{0, 1\}^{n \times n}$, define its associated *pattern cone*

$$\mathcal{K}(D) := \big\{ v \in \mathbb{R}^d : (2D - I) Xv \geq 0 \text{ (entrywise)} \big\}.$$

Then $v \in \mathcal{K}(D)$ if and only if $D(X, v) = D$ (up to measure-zero ties on $Xu = 0$). In particular, for $v \in \mathcal{K}(D)$ we have the identity

$$[Xv]_+ = D Xv, \tag{5}$$

where $[\cdot]_+$ is taken entrywise.

## A.3. From the convex program to a two-layer ReLU

Recall the sampled-pattern convex model in Section 3.1:

$$\min_{\{(v_i, w_i)\}_{i=1}^P} \ell\left(\sum_{i=1}^P D_i X (v_i - w_i), y\right) + \beta \sum_{i=1}^P \big(\|v_i\| + \|w_i\|\big) \quad \text{s.t.} \quad v_i, w_i \in \mathcal{K}(D_i). \tag{6}$$

Here $D_i \in \mathcal{D}_X$ are (sampled) activation patterns. In the multi-class case, take $v_i, w_i \in \mathbb{R}^{d \times K}$ with columns $(v_{i,1}, \ldots, v_{i,K})$ etc., and interpret $\|\cdot\|$ as either the block $\ell_{2,1}$ norm, $\|M\|_{2,1} = \sum_{k=1}^K \|M_{:,k}\|_2$, or the Frobenius norm.

## A.4. Variation norm and a computable certificate

We use the standard two-layer ReLU variation norm:

$$\|f\|_{\mathrm{var}} := \inf \left\{ \sum_{j=1}^m \|a_j\|_2 \|u_j\|_2 : f(h) = \sum_{j=1}^m a_j [u_j^\top h]_+, \ m \in \mathbb{N} \right\}.$$

The following result turns any feasible solution of (6) into an explicit upper bound on $\|f\|_{\mathrm{var}}$, therefore a Lipschitz certificate for the logits.

## Proof of Theorem 4.7

*Proof.* Using the representation in Proposition 4.6, build a two-layer network whose hidden units are the *columns* $\{v_{i,k}\}_{i,k}$ and $\{w_{i,k}\}_{i,k}$ with output weights $\{+e_k\}$ and $\{-e_k\}$ respectively. For each unit $(u, a)$ in this network, the atom cost is $\|a\|_2\|u\|_2 = \|u\|_2$ because $\|e_k\|_2 = 1$. Summing over units gives $\sum_{i,k} (\|v_{i,k}\|_2 + \|w_{i,k}\|_2) = \sum_i (\|v_i\|_{2,1} + \|w_i\|_{2,1})$, which upper bounds $\|f\|_{\mathrm{var}}$ by definition. For Frobenius penalties, $\sum_k \|M_{:,k}\|_2 \leq \sqrt{K}\|M\|_F$ yields the stated factor $\sqrt{K}$. $\qquad\square$

By Lemma 4.3, $\|f(h) - f(h')\|_\infty \leq \|f\|_{\mathrm{var}}\|h - h'\|_2$; combining with Theorem 4.4 yields the computable bounds

$$\mathrm{mar}(h + \delta, y) \geq \mathrm{mar}(h, y) - 2\,\widehat{\mathcal{B}}_{\mathrm{cvx}}^{(2,1)}\|\delta\|_2, \quad \mathrm{mar}(h + \delta, y) \geq \mathrm{mar}(h, y) - 2\sqrt{K}\,\widehat{\mathcal{B}}_{\mathrm{cvx}}^{\mathrm{F}}\|\delta\|_2,$$

depending on which penalty is used in the convex objective. If the encoder $E$ is $L_E$-Lipschitz, replace $\|\delta\|_2$ by $L_E\|x - x'\|_2$ to get the end-to-end certificate.

## A.5. AM–GM link to the nonconvex $\ell_2^2$ penalty

Consider the two-layer model $f(h) = \sum_{j=1}^m a_j[u_j^\top h]_+$ trained via the nonconvex penalty: $(\beta/2)\sum_{j=1}^m (\|a_j\|_2^2 + \|u_j\|_2^2)$. By AM–GM, $2\|a_j\|_2\|u_j\|_2 \leq \|a_j\|_2^2 + \|u_j\|_2^2$, hence

$$\|f\|_{\mathrm{var}} \leq \frac{1}{2}\sum_{j=1}^m (\|a_j\|_2^2 + \|u_j\|_2^2) = \frac{1}{\beta}\frac{\beta}{2}\sum_{j=1}^m (\|a_j\|_2^2 + \|u_j\|_2^2).$$

Therefore any solution of Eq. 2 yields the certificate

$$\|f\|_{\mathrm{var}} \leq \frac{1}{\beta}\mathcal{R}_{\ell_2^2} \quad\Rightarrow\quad \mathrm{mar}(h + \delta, y) \geq \mathrm{mar}(h, y) - \frac{2}{\beta}\mathcal{R}_{\ell_2^2}\|\delta\|_2.$$

Larger $\beta$ tightens the bound linearly.

## A.6. Details for the logit Lipschitz bound

Let $f(h) = \sum_j a_j[u_j^\top h]_+$. Since $t \mapsto [t]_+$ is 1-Lipschitz,

$$|f_k(h) - f_k(h')| = \Big|\sum_j a_{j,k}([u_j^\top h]_+ - [u_j^\top h']_+)\Big| \leq \sum_j |a_{j,k}|\,|u_j^\top(h - h')| \leq \sum_j \|a_j\|_2\|u_j\|_2\,\|h - h'\|_2.$$

Taking $\max_k$ and the infimum over all representations yields $\|f(h) - f(h')\|_\infty \leq \|f\|_{\mathrm{var}}\|h - h'\|_2$, which is the Lemma used in Theorem 4.4.

# B. Related Work Continues

**Training Data Scarcity.** Contemporary studies (Reitmaier et al., 2022) have identified lack of training data as a dominant challenge to improving performance in speech models. To address this, authors (Babirye et al., 2022) have worked on building partnerships to document valuable linguistic data by remotely engaging participants to record themselves, identifying more recording opportunities, and categorizing challenges of ASR in deeply multicultural communities. This has uncovered valuable implications for collaborations across ASR and Human Computer Interface (HCI) that advance important discussions and yielded more diverse speech datasets. However, this approach also brings up new questions on the ethics of analyzing community voice recordings through platforms such as WhatsApp (Barbosa & Milan, 2019), and is slow to provide clearly annotated training data from global low-resource languages.

**Certified Robustness in Deep Learning.** Ensuring model reliability under input perturbations is a critical area of study, which is typically addressed via empirical defenses like adversarial training (Madry et al., 2018). However, empirical defenses often do not provide formal guarantees. Research in certified robustness and information theory aim to provide provable bounds on classifier consistency (e.g., via randomized smoothing or Lipschitz analysis) (Cohen et al., 2019; Tsuzuku et al., 2018). In the audio domain, robustness is usually evaluated against environmental noise rather than dialectal shifts. By leveraging the specific geometry of our convex reformulation, we derive a variation-norm certificate that bounds the Lipschitz constant of the detection head directly. Unlike black-box certification methods, our approach yields a constructive certificate of margin stability derived explicitly from the optimization objective.

**GPU Accelerated JAX Methods.** The importance of GPU acceleration has fueled much of the success in contemporary optimization and deep learning. The strong parallelization and speed potential of this framework has been demonstrated in the optimization works of Shang et al. (2025), Du & He (2025), and Toutlini et al. (2025). This motivates us to solve the convex reformulation with a batched, multi-GPU ADMM solver (Boyd et al., 2011), yielding rapid convergence and efficient use of modern accelerators. As a result, our open-source implementation is easy to reproduce and plug into existing ASR pipelines or other encoder–decoder architectures. This design also aligns CLD with recent JAX-based convex and LLM systems such as CRONOS (Feng et al., 2024), and opens a clear path for future work on scaling convex language detection to larger encoders, cloud TPUs, and broader multilingual speech and multimodal agent application settings.

## C. Alternating Direction Method of Multipliers

The Alternating Direction Method of Multipliers (ADMM) is a divide-and-conquer optimization algorithm that has demonstrated significant prominence in solving large-scale convex optimization problems, particularly those involving decomposable objective functions and constraints.

**ADMM Background.** ADMM is an optimization framework designed to efficiently solve convex minimization problems that involve complex or non-smooth regularization terms. It works by breaking down a large global problem into a sequence of smaller, easier-to-solve local subproblems, which are then solved iteratively and linked together via Lagrange multipliers (the "multipliers" component) and a quadratic penalty term (the "augmented Lagrangian" component). This structure is ideal when the objective function is composed of several parts that apply to different subsets of variables, making the problem separable.

**Novelty and Significance.** In the context of Convex Language Detection (CLD), ADMM is critical and represents a significant novelty since it provides:

- **Tractability for Scale:** CLD is derived from a convex reformulation of a two-layer ReLU network. While theoretically sound, solving the resulting optimization problem (Eq. 2) can be extremely challenging for high-dimensional data, such as the extracted ASR hidden features. ADMM makes this problem tractable by allowing the optimization to be performed in a batched, multi-GPU parallel fashion (especially when implemented in JAX). ADMM is perfectly suited for solving such problems by isolating the non-smooth term in one subproblem (the proximal update).

- **Guaranteed Global Optimality:** By solving the convex program using a principled method like ADMM, the training process is guaranteed to converge to the unique global optimum. This eliminates the dependency on brittle hyperparameters (like learning rates) and avoids the local minima issues that plague traditional neural network training (e.g., the standard MLP baseline).

- **Novelty in ASR/Speech:** While ADMM is well-known in control and optimization (Boyd et al., 2011), we assert that its practical application as a core training method for complex deep learning reformulations on large-scale spoken dialogue systems remains highly novel. This innovation allows CLD to achieve massive compute efficiency (requiring 13x fewer TFLOPs than the standard vanilla-NN) and drastically reduced training time.

# D. Extended Experimental Results and Discussion

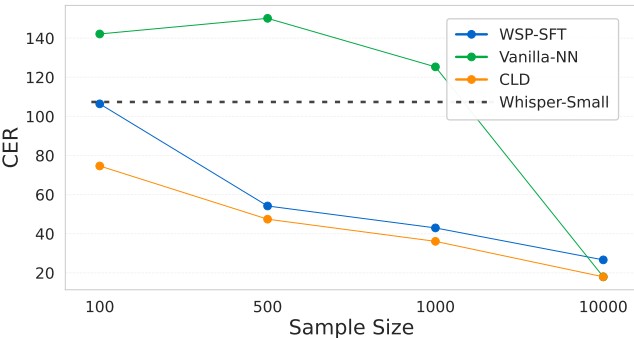

*Figure D.1.* Character error rates (CER) of Whisper-small default, Whisper-small fine-tuned, vanilla-NN, and CLD on binary classification between English and Chinese across training sample sizes of 100, 500, 1000, and 10 000.

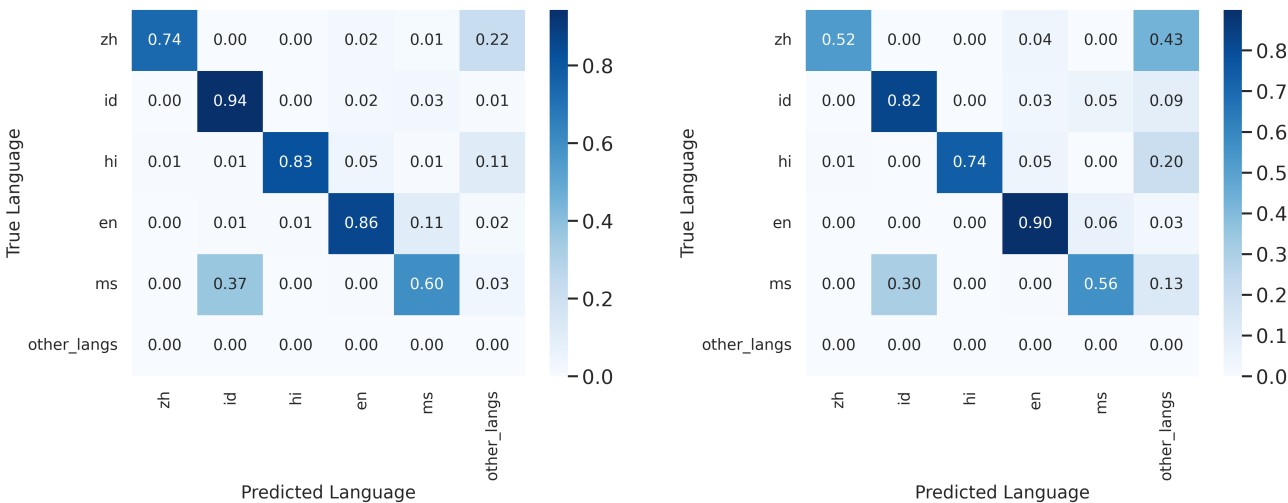

*Figure D.2.* Language Classification Performance between WSP-SFT (left) and WSP (right) on Whisper-small across English (en), Chinese (zh), Indonesian (id), Malaysian (ms), Hindi (hi), and other predicted languages.

## D.1. Classification Accuracy per Sample Size Ablation

*Table D.1.* Multi-dialect classification accuracy between English (EN) and Chinese (ZH) across 10 accents for (Whisper-baseline) WSP, WSP-SFT, vanilla-NN, and CLD at 100 samples per language and 20 samples per dialect.

| LANGUAGE - DIALECT | SIZE | CORRECTLY PREDICTED SAMPLES | | | | ACCURACY | | | |
|---|---|---|---|---|---|---|---|---|---|
| | | WSP | WSP-SFT | NN | CLD | WSP | WSP-SFT | NN | CLD |
| EN-HINDI | 190 | 176 | 93 | 0 | 185 | 0.9263 | 0.9263 | 0.0000 | **0.9737** |
| EN-MALAYSIAN | 215 | 136 | 137 | 5 | 213 | 0.6326 | 0.6372 | 0.0233 | **0.9907** |
| EN-SINGAPOREAN | 205 | 166 | 166 | 0 | 203 | 0.8098 | 0.8098 | 0.0000 | **0.9902** |
| EN-PAKISTANI | 189 | 182 | 182 | 10 | 186 | 0.9630 | 0.9630 | 0.0529 | **0.9841** |
| EN-AMERICAN | 204 | 195 | 195 | 4 | 203 | 0.9559 | 0.9559 | 0.0196 | **0.9951** |
| ZH-MIN DONG / FUZHOU | 71 | 7 | 7 | 64 | 71 | 0.0986 | 0.0986 | 0.9014 | **1.0000** |
| ZH-PU-XIAN | 216 | 32 | 33 | 193 | 213 | 0.1481 | 0.1528 | 0.8935 | **0.9861** |
| ZH-HONG KONG | 184 | 121 | 121 | 172 | 175 | 0.6576 | 0.6576 | 0.9348 | **0.9511** |
| ZH-TAIWANESE | 181 | 176 | 175 | 174 | 181 | 0.9724 | 0.9669 | 0.9613 | **1.0000** |
| ZH-MAINLAND | 205 | 187 | 187 | 195 | 199 | 0.9122 | 0.9122 | 0.9512 | **0.9707** |
| TOTAL | 1860 | 1378 | 1286 | 817 | 1829 | 0.7077 | 0.7102 | 0.4738 | **0.9742** |

*Table D.2.* Multi-dialect classification accuracy between English (EN) and Chinese (ZH) across 10 accents for (Whisper-baseline) WSP, WSP-SFT, vanilla-NN, and CLD at 1000 samples per language and 200 samples per dialect.

| LANGUAGE - DIALECT | SIZE | CORRECTLY PREDICTED SAMPLES | | | | ACCURACY | | | |
|---|---|---|---|---|---|---|---|---|---|
| | | WSP | WSP-SFT | NN | CLD | WSP | WSP-SFT | NN | CLD |
| EN-HINDI | 190 | 171 | 175 | 190 | 185 | 0.9000 | 0.9211 | **1.0000** | 0.9737 |
| EN-MALAYSIAN | 215 | 123 | 115 | 215 | 214 | 0.5714 | 0.5349 | **1.0000** | 0.9953 |
| EN-SINGAPOREAN | 205 | 164 | 154 | 205 | 203 | 0.8000 | 0.7512 | **1.0000** | 0.9902 |
| EN-PAKISTANI | 189 | 178 | 180 | 189 | 187 | 0.9444 | 0.9524 | **1.0000** | 0.9894 |
| EN-AMERICAN | 204 | 178 | 195 | 204 | 203 | 0.9444 | 0.9559 | **1.0000** | 0.9951 |
| ZH-MIN DONG / FUZHOU | 71 | 19 | 30 | 15 | 69 | 0.2727 | 0.4225 | 0.2113 | **0.9718** |
| ZH-PU-XIAN | 216 | 46 | 82 | 56 | 216 | 0.2143 | 0.3796 | 0.2593 | **1.0000** |
| ZH-HONG KONG | 184 | 124 | 143 | 57 | 182 | 0.6667 | 0.7772 | 0.3098 | **0.9891** |
| ZH-TAIWANESE | 181 | 159 | 181 | 39 | 181 | 0.8824 | **1.0000** | 0.2155 | **1.0000** |
| ZH-MAINLAND | 205 | 178 | 192 | 55 | 204 | 0.8696 | 0.9366 | 0.2683 | **0.9951** |
| TOTAL | 1860 | 1314 | 1447 | 1225 | 1844 | 0.7066 | 0.7632 | 0.6586 | **0.9914** |

*Table D.3.* Multi-dialect classification accuracy between English (EN) and Chinese (ZH) across 10 accents for (Whisper-baseline) WSP, WSP-SFT, vanilla-NN, and CLD at 10 000 samples per language and 2000 samples per dialect.

| LANGUAGE - DIALECT | SIZE | CORRECTLY PREDICTED SAMPLES | | | | ACCURACY | | | |
|---|---|---|---|---|---|---|---|---|---|
| | | WSP | WSP-SFT | NN | CLD | WSP | WSP-SFT | NN | CLD |
| EN-HINDI | 190 | 176 | 180 | 190 | 180 | 0.9263 | 0.9474 | **1.0000** | 0.9474 |
| EN-MALAYSIAN | 215 | 136 | 135 | 215 | 194 | 0.6326 | 0.6279 | **1.0000** | 0.9023 |
| EN-SINGAPOREAN | 205 | 166 | 166 | 198 | 195 | 0.8098 | 0.8098 | **0.9659** | 0.9512 |
| EN-PAKISTANI | 189 | 182 | 185 | 189 | 183 | 0.9630 | 0.9788 | **1.0000** | 0.9683 |
| EN-AMERICAN | 204 | 195 | 197 | 197 | 194 | 0.9559 | **0.9657** | **0.9657** | 0.9509 |
| ZH-MIN DONG / FUZHOU | 71 | 7 | 50 | 71 | 71 | 0.0986 | 0.7042 | **1.0000** | 1.0000 |
| ZH-PU-XIAN | 216 | 32 | 198 | 214 | 216 | 0.1481 | 0.9167 | 0.9907 | 1.0000 |
| ZH-HONG KONG | 184 | 121 | 170 | 173 | 184 | 0.6576 | 0.9239 | 0.9402 | 1.0000 |
| ZH-TAIWANESE | 181 | 176 | 181 | 181 | 181 | 0.9724 | **1.0000** | **1.0000** | 1.0000 |
| ZH-MAINLAND | 205 | 187 | 195 | 205 | 205 | 0.9122 | 0.9512 | **1.0000** | 1.0000 |
| TOTAL | 1860 | 1378 | 1657 | 1833 | 1803 | 0.7077 | 0.8909 | **0.9855** | 0.9694 |

## D.2. Impact of Hyperparameter Tuning on Vanilla NN Performance

This appendix provides the detailed breakdown of performance changes following hyperparameter tuning for the Vanilla NN classifier across various sample sizes and base models.

*Table D.4.* Multi-dialect classification accuracy between English (EN) and Chinese (ZH) for vanilla-NN following hyperparameter tuning across 100, 500, 1000, and 10000 samples. Change in accuracy from the non-tuned model is shown in parentheses.

| DIALECT | SIZE | 100 SAMPLES | 500 SAMPLES | 1000 SAMPLES | 10000 SAMPLES |
|---|---|---|---|---|---|
| EN-HINDI | 190 | 0.0000 (-3.7%) | 1.0000 (0.0%) | 1.0000 (0.0%) | 1.0000 (+1.6%) |
| EN-MALAYSIAN | 215 | 0.0233 (-0.9%) | 0.9953 (-0.5%) | 1.0000 (0.0%) | 1.0000 (0.0%) |
| EN-SINGAPOREAN | 205 | 0.0000 (-1.5%) | 0.9756 (-2.4%) | 1.0000 (0.0%) | 0.9659 (-2.9%) |
| EN-PAKISTANI | 189 | 0.0529 (0.0%) | 1.0000 (0.0%) | 1.0000 (0.0%) | 1.0000 (+0.5%) |
| EN-AMERICAN | 204 | 0.0196 (-3.9%) | 0.9755 (-2.5%) | 1.0000 (0.0%) | 0.9657 (-3.4%) |
| ZH-MIN DONG | 71 | 0.9014 (-4.2%) | 0.2535 (-4.2%) | 0.2113 (+5.6%) | 1.0000 (+1.4%) |
| ZH-PU-XIAN | 216 | 0.8935 (-5.1%) | 0.2037 (0.0%) | 0.2593 (+4.2%) | 0.9907 (-0.5%) |
| ZH-HONG KONG | 184 | 0.9348 (-1.6%) | 0.0000 (-2.7%) | 0.3098 (+2.2%) | 0.9402 (-0.5%) |
| ZH-TAIWANESE | 181 | 0.9613 (-3.3%) | 0.0884 (+2.2%) | 0.2155 (+3.3%) | 1.0000 (0.0%) |
| ZH-MAINLAND | 205 | 0.9512 (-2.9%) | 0.0878 (+2.4%) | 0.2683 (+4.9%) | 1.0000 (+3.9%) |
| **TOTAL** | 1860 | 0.4738 (-2.7%) | 0.5580 (-0.8%) | 0.6264 (+2.0%) | 0.9862 (0.0%) |

*Table D.5.* WER, CER, and Detection Accuracy metrics between WSP, WSP-L, and MMS-1B for vanilla-NN following hyperparameter tuning. Change in accuracy from the non-tuned model is shown in parentheses.

| BASE MODEL | TUNED ACC | TUNED WER | TUNED CER |
|---|---|---|---|
| WSP | 0.7737 (+1.6%) | 53.84 (-4.74) | 34.52 (-2.61) |
| WSP-L | 0.9605 (-0.9%) | 29.25 (+0.42) | 15.99 (+0.33) |
| MMS-1B | 0.8612 (+0.1%) | 48.26 (-0.02) | 23.64 (-0.02) |

## D.3. Low-Resource Dialects in High-Resource Languages.

Although large-scale audio–text datasets have driven recent advances in ASR, high-quality speech data remains costly to collect and difficult to annotate, particularly for languages with substantial dialectal variation. Even the seminal corpora underlying modern ASR models (e.g., the reported 680 000+ hours used to train Whisper by Radford et al. (2023)) concentrate heavily on a small set of dominant high-resource speech within English and Mandarin, leaving many regional dialects and accented speech varieties underrepresented. This imbalance yields a critical performance gap: speakers of dialects such as Singaporean English ("Singlish") or regional Mandarin varieties experience significantly degraded transcription quality. For example, Table E.6 shows Whisper achieving 100% accuracy on Midwestern English but only 61.4% on Malaysian-accented English, despite both being the same language. These disparities highlight that even widely adopted ASR models frequently misidentify dialectal speech, limiting accessibility, reliability and trust for millions of users.

# E. Language classification accuracy of Baseline Whisper ASR

*Table E.6.* Whisper-small's default language detection accuracy by language and accent.

| LANGUAGE | ACCENT | SAMPLES | CORRECT | ACCURACY (%) |
|---|---|---|---|---|
| **ID** | | | | |
| | BETAWI | 505 | 451 | 89.3 |
| | JAVANESE | 182 | 163 | 89.6 |
| | JAWA TENGAH | 228 | 206 | 90.4 |
| | SURAKARTA | 221 | 200 | 90.5 |
| | BINDENG | 221 | 200 | 90.5 |
| | TIONGHOA | 221 | 200 | 90.5 |
| | MEDHOK | 224 | 203 | 90.6 |
| **EN** | | | | |

| LANGUAGE | ACCENT | SAMPLES | CORRECT | ACCURACY (%) |
|---|---|---|---|---|
| | MALAYSIAN ENGLISH | 1000 | 614 | 61.4 |
| | FILIPINO | 1000 | 745 | 74.5 |
| | SINGAPOREAN ENGLISH | 1000 | 752 | 75.2 |
| | ZIMBABWE | 1000 | 831 | 83.1 |
| | SOUTHERN AFRICAN (SOUTH AFRICA, NAMIBIA) | 1000 | 831 | 83.1 |
| | WELSH ENGLISH | 1000 | 918 | 91.8 |
| | SCANDINAVIAN | 1000 | 952 | 95.2 |
| | PAKISTAN | 1000 | 967 | 96.7 |
| | INDIA AND SOUTH ASIA (INDIA, SRI LANKA) | 1000 | 967 | 96.7 |
| | SCOTTISH ENGLISH | 1000 | 968 | 96.8 |
| | LANCASHIRE ENGLISH | 1000 | 970 | 97.0 |
| | LIVERPOOL ENGLISH | 1000 | 970 | 97.0 |
| | ENGLAND ENGLISH | 1000 | 982 | 98.2 |
| | AUSTRALIAN ENGLISH | 1000 | 984 | 98.4 |
| | UNITED STATES ENGLISH | 1000 | 985 | 98.5 |
| | NEW ZEALAND ENGLISH | 1000 | 987 | 98.7 |
| | HONG KONG ENGLISH | 1000 | 988 | 98.8 |
| | GERMAN ENGLISH | 1000 | 996 | 99.6 |
| | NON NATIVE SPEAKER | 1000 | 996 | 99.6 |
| | IRISH ENGLISH | 1000 | 996 | 99.6 |
| | CANADIAN ENGLISH | 1000 | 999 | 99.9 |
| | NORTHERN IRISH | 1000 | 999 | 99.9 |
| | LOW | 1000 | 999 | 99.9 |
| | DEMURE | 1000 | 999 | 99.9 |
| | MIDWESTERN | 1000 | 1000 | 100.0 |
| | TRANSATLANTIC ENGLISH | 1000 | 1000 | 100.0 |
| **ZH** | | | | |
| | CDO | 1000 | 103 | 10.3 |
| | CPX | 1000 | 111 | 11.1 |
| | NAN-TW | 1000 | 545 | 54.5 |
| | HK | 1000 | 905 | 90.5 |
| | ZH | 1000 | 910 | 91.0 |
| | TW | 1000 | 987 | 98.7 |
| **HI** | | | | |
| | KASHMIRI | 320 | 185 | 57.8 |
| | BODO | 380 | 270 | 71.1 |
| | MALAYALAM | 365 | 271 | 74.2 |
| | PUNJABI | 236 | 178 | 75.4 |
| | URDU | 131 | 103 | 78.6 |
| | TELUGU | 474 | 377 | 79.5 |
| | TAMIL | 182 | 145 | 79.7 |
| | HINDI | 575 | 468 | 81.4 |
| | SINDHI | 141 | 116 | 82.3 |
| | ODIA | 374 | 311 | 83.2 |
| | KANNADA | 313 | 268 | 85.6 |
| | GUJARATI | 299 | 257 | 86.0 |
| | ASSAMESE | 262 | 226 | 86.3 |
| | KONKANI | 422 | 364 | 86.3 |
| | NEPALI | 359 | 311 | 86.6 |
| | BENGALI | 418 | 366 | 87.6 |
| | DOGRI | 219 | 194 | 88.6 |
| | MAITHILI | 287 | 255 | 88.9 |
| | MARATHI | 395 | 366 | 92.7 |
| **MS** | | | | |
| | MSI | 1000 | 386 | 38.6 |
| | MS | 1000 | 934 | 93.4 |

## F. Qualitative Human Case Study

| METHOD | TOTAL TEST PROMPTS | WRONG LANGUAGE TRANSCRIBED | WORD ERRORS IN TRANSCRIPTION |
|---|---|---|---|
| *Default* | | | |
| EN | 595 | 59 | – |
| ZH | 300 | 148 | – |
| *vanilla-NN* | | | |
| EN | 450 | 22 | 81 |
| ZH | 450 | 5 | 14 |
| *CLD (ours)* | | | |
| EN | 450 | 12 | 26 |
| ZH | 450 | 2 | 14 |

*Table F.7.* Illustrative human case-study comparison on Whisper-small using the Default, vanilla-NN, and CLD language classifiers. The study involved five participants in Singapore for English (EN) and ten participants in southeastern China for Mandarin (ZH).

## G. Hyperparameter Selection and Hardware Setup

In all models, we tune the hyperparameter by choosing the best configuration based on performance over the validation set. We split using a 80-10-10 train, test, validation split.

**CLD parameters.** Since one of the features of CLD is its lack of brittle hyperparameters, we implement design choices with all suggested default parameters presented from the subroutines utilized. For binary classification, we set rank $= 20$, $\beta = 10^{-3}$, $\rho = 10^{-4}$, $\gamma$-ratio $= 1$, ADMM iterations $= 6$, PCG iterations $= 32$, with neuron count $= 10$. For multiclass classification, we use the same settings but set the neuron count to 32.

**Neural Network Baseline.** The NN baseline was trained using AdamW with hyperparameters selected via grid search over learning rates $\{10^{-4}, 3 \times 10^{-4}, 10^{-3}, 3 \times 10^{-3}\}$, weight decays $\{0, 10^{-5}, 10^{-4}, 10^{-3}\}$, and epoch counts $\{5, 10, 20\}$. The best configuration was selected based on validation accuracy.

**Linear SVM Baseline.** We used a LinearSVC classifier with a maximum of 5000 iterations. The inverse regularization parameter $C$ was selected via grid search over $\{10^{-2}, 10^{-1}, 1, 10, 100\}$. Input embeddings were standardized with zero mean and unit variance prior to training.

**Kernel SVM Baseline.** We used an SVC classifier with input embeddings standardized prior to training. Hyperparameters were selected via grid search over kernel types $\{\texttt{rbf}, \texttt{poly}, \texttt{sigmoid}\}$, regularization values $C \in \{0.1, 1, 10, 100\}$, and kernel coefficients $\gamma \in \{\texttt{scale}, \texttt{auto}\}$. The polynomial kernel was evaluated at degree 3. The best configuration was selected based on validation accuracy.

$k$-**Nearest Neighbours Baseline.** We used a $k$-NN classifier with input embeddings standardized prior to training. Hyperparameters were selected via grid or random search over $k \in \{3, 5, 7, 11, 15, 21\}$, distance metrics $\{\texttt{euclidean}, \texttt{cosine}, \texttt{manhattan}\}$, and weighting schemes $\{\texttt{uniform}, \texttt{distance}\}$. The best configuration was selected based on validation accuracy.

**Hardware.** All experiments were conducted on four NVIDIA A100-SXM4 GPUs, each with 40 GB of memory.

