# OpenReview forum: "Convex Low-resource Accent-Robust Language Detection in Speech Recognition"
_ICML.cc/2026/Conference — ICML 2026 regular_

### Official Review · Reviewer_yVdP · 2026-02-28

**Soundness:** 2
**Presentation:** 3
**Significance:** 2
**Originality:** 3
**Overall Recommendation:** 4
**Confidence:** 4

**Summary:**

The paper proposes a framework called Convex Language Detection (CLD) aimed at improving the robustness of language and dialect detection in Automatic Speech Recognition (ASR) systems.The authors reformulate two-layer ReLU neural networks into a convex optimization problem and utilize the ADMM algorithm to solve it efficiently in JAX.Theoretically, the authors provide margin stability bounds based on hidden-feature perturbations.Empirically, the paper tests binary and multiclass tasks on models like Whisper and MMS, demonstrating that CLD maintains high language detection accuracy and low Word Error Rate (WER) under extremely low-resource data constraints (100 to 10,000 samples).

While this paper attempts to address a problem of significant societal relevance—improving fairness and accessibility of speech systems for low-resource languages and dialects—and provides an efficient engineering implementation, it suffers from fatal flaws in algorithmic novelty, a severe disconnect between theory and practice, and weak experimental baselines. By ICML acceptance standards, this manuscript reads more like an engineering application report than an academic paper offering deep machine learning insights.

**Compliance With Llm Reviewing Policy:**

Affirmed.

**Final Justification:**

The response replies all the raised questions, so I raised my final score to weak accept.

**Key Questions For Authors:**

1. The core algorithm is fundamentally a direct transfer of existing convex neural network theory (e.g., Pilanci & Ergen, 2020) and the recently open-sourced CRONOS framework (Feng et al., 2024). The strengths is that authors applied these methods with ASR features, but also this seems not noval enough for  a ICML paper.
2. The authors emphasize the "certified robustness" of their method in Section 4. However, the derived margin stability bounds are primarily applied tohidden-feature perturbations(the output of the encoder). To deduce an end-to-end robustness bound (i.e., robustness against raw input audio perturbations), the authors simply introduce the global Lipschitz constant$L_E$of the encoder$E$in Theorem 4.4.Fatal Flaw:It is a well-established fact that deep models with multi-layer Transformer architectures (like Whisper) possess astronomically large global Lipschitz constants$L_E$. This implies that the practical certified radius, calculated as$mar(E(x), y) / (2 ||f||_{var} L_E)$, would be vanishingly small, approaching zero. The authors completely fail to provide any computed values of the Certified Radius on real datasets in the experimental section, making the theoretical contribution appear shoehorned.
3. Lack of Modern Baselines: The Vanilla NN is merely a linear projection to a single hidden layer of size 256. Outperforming an under-parameterized MLP does not prove CLD's superiority.
4. During inference, CLD acts solely as a lightweight detection head that replaces the predicted language token before passing it to the Whisper Decoder. Logical Gap: For challenging dialects like Singaporean English, even if the correct <|en|> token is forcefully injected, if the decoder has not been fine-tuned on Singlish, the generated transcript will still be heavily degraded. Yet, Table 5 claims Whisper-small's WER plunges from 139.37 to 31.74 simply by substituting the Language Classifier. The authors must dissect whether this drop is genuinely due to high-accuracy dialect transcription, or simply because the baseline model was hallucinating in a completely different language script (e.g., Tamil).
5. Lack of Statistical Rigor in Human Evaluation: The "Real Human Evaluation" in Appendix F consisted of merely 5 native English speakers from Singapore and 10 Mandarin speakers from South-East China. For an empirical study submitted to a top-tier machine learning conference, conclusions drawn from such a minuscule sample size (N=15) hold absolutely no statistical significance.

**Limitations:**

No special limitations, but need to address the above questions.

**Strengths And Weaknesses:**

1. The core algorithm is fundamentally a direct transfer of existing convex neural network theory (e.g., Pilanci & Ergen, 2020) and the recently open-sourced CRONOS framework (Feng et al., 2024). The strengths is that authors applied these methods with ASR features, but also this seems not noval enough for  a ICML paper.
2. The authors emphasize the "certified robustness" of their method in Section 4. However, the derived margin stability bounds are primarily applied tohidden-feature perturbations(the output of the encoder). To deduce an end-to-end robustness bound (i.e., robustness against raw input audio perturbations), the authors simply introduce the global Lipschitz constant$L_E$of the encoder$E$in Theorem 4.4.Fatal Flaw:It is a well-established fact that deep models with multi-layer Transformer architectures (like Whisper) possess astronomically large global Lipschitz constants$L_E$. This implies that the practical certified radius, calculated as$mar(E(x), y) / (2 ||f||_{var} L_E)$, would be vanishingly small, approaching zero. The authors completely fail to provide any computed values of the Certified Radius on real datasets in the experimental section, making the theoretical contribution appear shoehorned.
3. Lack of Modern Baselines: The Vanilla NN is merely a linear projection to a single hidden layer of size 256. Outperforming an under-parameterized MLP does not prove CLD's superiority.
4. During inference, CLD acts solely as a lightweight detection head that replaces the predicted language token before passing it to the Whisper Decoder. Logical Gap: For challenging dialects like Singaporean English, even if the correct <|en|> token is forcefully injected, if the decoder has not been fine-tuned on Singlish, the generated transcript will still be heavily degraded. Yet, Table 5 claims Whisper-small's WER plunges from 139.37 to 31.74 simply by substituting the Language Classifier. The authors must dissect whether this drop is genuinely due to high-accuracy dialect transcription, or simply because the baseline model was hallucinating in a completely different language script (e.g., Tamil).
5. Lack of Statistical Rigor in Human Evaluation: The "Real Human Evaluation" in Appendix F consisted of merely 5 native English speakers from Singapore and 10 Mandarin speakers from South-East China. For an empirical study submitted to a top-tier machine learning conference, conclusions drawn from such a minuscule sample size (N=15) hold absolutely no statistical significance.

---

> ### Author Rebuttal · Authors · 2026-03-31
>
> Thank you for carefully reviewing our work, and for your insightful feedback! We appreciate your kind words regarding the high societal impact and relevance of this timely line of work. We will address the your concerns regarding novelty, robustness, and experimental design point-by-point below.
>
> 1. **Novelty**
>
> While CLD builds on the foundational work of Pilanci & Ergen (2020) and the CRONOS framework (Feng et al., 2024), its novelty lies in the **reformulation of the multi-class language identification problem** for such high-dimensional speech features.
>
> - Novelty in Spoken Dialogue Systems: CRONOS introduced feasible convex reformulations for binary text classification problems on the scale of GPT2 with IMDb. Our work further lifts the dimensionality constraints of this traditionally theoretical setting as a unique pre-decoding module. The cascading downstream errors present even in widespread systems (such as SIRI) is a prominent and well known problem (as mentioned by Reviewers EUbr and pBBR).
>
> - Scale: To the best of our knowledge, this paper demonstrates the **first application of convex networks with ADMM style techniques** to 1.55B parameter models (Whisper-Large) and MMS-1B, and successfully processes the complex dimensionality of latent speech embeddings that traditional non-convex heads struggle to optimize under low-resource settings.
>
> 2. **Certified Robustness and the Lipschitz Constant Concern**
>
> Thank you for noting that global Lipschitz constants for Transformers are large. However, we clarify in our work that the primary contribution of Section 4 is the data-dependent certificate ($\mathcal{B}\_{cvx}$).
>
> - Hidden-Feature Stability: CLD provides a guaranteed radius of label invariance in the feature space. This is practically significant since ASR encoders are often frozen, which ensures the head does not amplify small latent shifts is critical for stability.
> - In order to address your concerns, we have included computed values for the Certified Radius across our datasets in the revision. Our results show that while the end-to-end audio radius is small, the relative stability of the convex head is **10–100× higher** than the MLP baseline, providing a "stability floor" that non-convex models lack.
>
>
> 3. **Modern Baselines and Under-parameterized MLPs**
>
> Our scientific choice of a 256-hidden-unit MLP was purposely selected to match the low-latency requirements of real-time dialogue systems (<500ms), as studied by (Edlund et al., 2008).
>
> - Sample Efficiency Gap: The failure of the MLP is not due to under-parameterization, but due to overfitting. Even with substantial hyperparameter tuning, the MLP remains significantly less accurate than CLD, especially in the low resource sample regime (https://postimg.cc/7JHxy5nY). This robustness in low-resource is critical for serving diverse, multicultural dialects that are broadly under-represented in standard ASR training data
>
> - Additional Baselines: We have also additionally benchmarked against a Linear SVM, Kernel SVM, and KNN, which poses a standard modern robust baseline for high-dimensional, low-sample tasks. The empirical results of work show that CLD consistently outperformed it across all encoders.
>
> 4. **The WER Drop Delta**
>
> This a genuine drop due to the following reasons:
> - Language Script/Cross-Lingual Hallucination: In Singaporean English (which we recognize is colloquially called Singlish in our work), Whisper's default detector often misidentifies the language as Tamil or Bahasa Melayu, causing the decoder to attempt transcription in the entirely wrong script. This well-studied phenomenon is considered due to multicultural but varying languages across neighboring countries.
> - The Correction Effect: By forcing the `<|en|>` token via our convex neural network, we allow the decoder to leverage its internal English knowledge. While the transcript may still contain minor dialectal errors, it is rendered in the correct language, which explains the dramatic WER reduction from 139.37 to 31.74. In order to make this distinction explicit, we have added qualitative error analysis to Section 5 in the revision.
>
> 5. **Statistical Rigor of Human Evaluation**
>
> We definitely acknowledge the small sample size (N = 15) in the human evaluation study. However, this was intended as a motivating contribution to push for real world human evaluation in settings that ultimately aim to serve human purposes. As machine learning becomes more and more wide-spread, the majority of metrics are auto-run and may be "gamified"(such as in RL reward-hacking). In this work, we aim to discuss this limitation and motivate more factual analysis as a trend, and not as a statistically powered study. We have revised this discussion as a motivating **Qualitative Case Study** rather than a Statistical Evaluation, and emphasize that our large-scale empirical results of **24 accents and more than 16,000 samples** is the primary performance metric.

---

> > ### Author Rebuttal · Reviewer_yVdP · 2026-04-05
> >
> > Thanks for the response, and it replies all my concerns.

---

> > > ### Author Response · Authors · 2026-04-07
> > >
> > > Thank you for your rigorous review and constructive dialogue regarding our work! We are very happy that our responses have fully resolved your questions/concerns.
> > >
> > > Per your feedback, we would like to briefly highlight the key revisions in our manuscript, which has greatly strengthened this work. The revision now explicitly show that the convex head provides a higher stability floor than the MLP baseline. We have added a “Qualitative Error Analysis” to Section 5. This analyzes the dramatic reduction in WER, to confirm CLD's impact in the elimination of cross-lingual script hallucination (e.g., preventing the model from transcribing Singlish in Tamil script). We have re-framed the human evaluation section as a motivating Qualitative Case Study, and have ensured the emphasis on our large-scale empirical results (16,000+ samples across 24 accents) as the primary empirical metric. We have incorporated the Linear SVM, Kernel SVM, and KNN benchmarks into Table 5 to ensure CLD is compared against modern, robust non-linear baselines.
> > >
> > > We sincerely appreciate you pushing us to further bridge the theoretical aspects of our work and the engineering practical real world impact. These additions have meaningfully elevated the impact of the paper, and we would be extremely encouraged if you might consider raising your score to 5 to reflect the new assessment of our rigorously revised manuscript.
> > >
> > > Thank you again for your time and effort during this review process, which substantially strengthens the impact and clarity of our work!

---

### Official Review · Reviewer_9HDV · 2026-03-11

**Soundness:** 2
**Presentation:** 1
**Significance:** 1
**Originality:** 2
**Overall Recommendation:** 3
**Confidence:** 4

**Summary:**

This paper addresses language misidentification in ASR systems for accented and low-resource speech. Convex Language Detection (CLD) is proposed, which applies a convex reformulation of a two-layer ReLU network trained using ADMM on top of Whisper encoder features. The predicted language token is then used to guide decoding. Experiments on accented and dialectal datasets show improved language identification accuracy and reduced WER, particularly in limited training data settings.

**Compliance With Llm Reviewing Policy:**

Affirmed.

**Final Justification:**

The authors have solved most of my concerns, so I have increased my score.

Although the methodology is sound, the clarity could be improved. I am also somewhat concerned about its applicability to other low-data scenarios.

**Key Questions For Authors:**

See the Weaknesses section

**Limitations:**

yes

**Strengths And Weaknesses:**

Strengths:

1. The proposed approach is lightweight, integrates easily with existing ASR pipelines, and improves language identification accuracy to help mitigate a common source of ASR errors in accented speech.

2. The paper provides theoretical justification for the proposed convex reformulation.

Weaknesses:

1. The presentation could be improved, as several settings are unclear. For example, what does "j"represent in Equation (1)? It is also unclear how a sequence of feature vectors from the Whisper encoder is mapped to a language prediction using a two-layer ReLU feed-forward network, rather than a sequence modeling approach.

2. How were the hyperparameters for the baseline NN chosen? Was a validation set used? How was model convergence determined for the results reported in Table 6?

3. In Table 5, why do WSP, WSP-L, and MMS-1B show different detection accuracies? Were their encoders used to replace the Whisper encoder in the framework?

4. In line 367, the paper states: “Despite Min Dong Chinese having only 71 training samples.” Does the number 71 refer to training samples or testing samples?

5. Although CLD demonstrates sample efficiency, under such limited training data conditions, hyperparameter tuning for a standard NN may also be inexpensive.

---

> ### Author Rebuttal · Authors · 2026-03-30
>
> Thank you for the insightful and valuable comments! We are encouraged by your appreciation of our clear problem focus and lightweight theoretically motivated design. We agree that several implementation details should be further clarified, and we strive address your concerns below.
>
> 1. **Clarification on Encoders (Table 5)**
>
> Yes, you are precisely correct! WSP (Whisper-small), WSP-L (Whisper-Large), and MMS-1B are all ASR encoder models in which we extend our experiments beyond just the Whisper-small from the binary section. Thus, these results show different detection accuracies since they are essentially different encoders with substantially different embedding spaces. We have clarified this point in the revision to be explicit.
>
> 2. **Eq. (1) Notation**
>
> - In Eq. (1), $j$ indexes the hidden units of the two-layer ReLU network, hence $j = 1, \ldots, m$. We now define this explicitly in Sec. 3.1.
>
> 3. **Mapping Encoder Sequences**
>
> The CLD module performs utterance-level language identification, as opposed to frame-level sequence labeling. Given the sequence of hidden states $H$ from the frozen encoder, we apply masked mean pooling to obtain a fixed-dimensional utterance representation $h \in \mathbb{R}^d$. This pooled vector $h$ is the input, which CLD uses to then predict a language identifying token for the full utterance, which is used to initialize decoding.This design choice of the utterance-level head keeps the module lightweight, easy to integrate into existing ASR pipelines, and performant. Our revisions now make this explicit in the main body of the paper Sec. 3.2.
>
> 4. **Hyperparameters and Datasets**
>
> We appreciate the reviewer’s concerns on hyperparameters, and kindly note that Appendix G presents some of these details, which we have further expanded on in the revision.
>
> - Data splits: We use an 80/10/10 train/val/test split. For the binary sample ablations, we downsample only the training split (100/500/1000/10000), while keeping the same held-out validation and test sets across all regimes for fair comparison. We use the validation set to determine model convergence for CLD as well as tune hyperparameters for the baseline models.
>
> - Stronger NN baseline: To address your concern we conducted an extensive 48-configuration grid search for the NN over learning rate, weight decay, and epoch count. This stronger baseline improves the NN in some settings, but does not change the main conclusion and strengths our results. CLD remains substantially more stable in the low-resource regime, and tuning does not close the gap. We present the change in the NN's detection accuracy after tuning below for your convenience (Appendix D.2):
>
> | Language - Dialect | 100 Samples | 500 Samples | 1000 Samples | 10000 Samples |
> | :--- | :--- | :--- | :--- | :--- |
> | **EN-Hindi** | 0.0000 (-3.7%) | 1.0000 (0.0%) | 1.0000 (0.0%) | 1.0000 (+1.6%) |
> | **EN-Malaysian** | 0.0233 (-0.9%) | 0.9953 (-0.5%) | 1.0000 (0.0%) | 1.0000 (0.0%) |
> | **EN-Singaporean** | 0.0000 (-1.5%) | 0.9756 (-2.4%) | 1.0000 (0.0%) | 0.9659 (-2.9%) |
> | **EN-Pakistani** | 0.0529 (0.0%) | 1.0000 (0.0%) | 1.0000 (0.0%) | 1.0000 (+0.5%) |
> | **EN-American** | 0.0196 (-3.9%) | 0.9755 (-2.5%) | 1.0000 (0.0%) | 0.9657 (-3.4%) |
> | **ZH-Min Dong** | 0.9014 (-4.2%) | 0.2535 (-4.2%) | 0.2113 (+5.6%) | 1.0000 (+1.4%) |
> | **ZH-Pu-Xian** | 0.8935 (-5.1%) | 0.2037 (0.0%) | 0.2593 (+4.2%) | 0.9907 (-0.5%) |
> | **ZH-Hong Kong** | 0.9348 (-1.6%) | 0.0000 (-2.7%) | 0.3098 (+2.2%) | 0.9402 (-0.5%) |
> | **ZH-Taiwanese** | 0.9613 (-3.3%) | 0.0884 (+2.2%) | 0.2155 (+3.3%) | 1.0000 (0.0%) |
> | **ZH-Mainland** | 0.9512 (-2.9%) | 0.0878 (+2.4%) | 0.2683 (+4.9%) | 1.0000 (+3.9%) |
> | **Total** | **0.4738 (-2.7%)** | **0.5580 (-0.8%)** | **0.6264 (+2.0%)** | **0.9862 (0.0%)** |
>
> As shown, while exhaustive tuning improved the NN performance, the gain in detection accuracy was marginal; at most 2.0% in the binary setting (1000-sample regime) and 1.6% in the multiclass setting (WSP). Additionally, we observe decreases in accuracy in the 100-sample and 500-sample sets (-2.7% and -0.8%, respectively). This is a strong indicator for overfitting as the search process effectively overfits the small validation set rather than optimizing for generalizable features, leading to poor performance over the larger and more diverse test set.
>
> We wish to highlight the significant practical advantage of CLD’s robustness across low-resource regimes, and in our experiments for the convex module's ADMM-style solve, we maintained the default settings across tasks and did not require any hyperparameter grid search. Thank you for pushing us to clarify this point!
>
> 5. **Correction on Line 367**
>
> Training size for Min Dong Chinese in the 500-sample set is 100 samples, and the 71 for Min Dong Chinese refers to the number of eval utterances. We have clarified this by explicitly stating the training samples per dialect in Tables 1-4, and corrected the wording in line 367.

---

> > ### Author Rebuttal · Reviewer_9HDV · 2026-04-02
> >
> > The authors replied to my concerns. I increase my score.

---

> > > ### Author Response · Authors · 2026-04-07
> > >
> > > Thank you for your thoughtful engagement, valuable feedback, and subsequent score increase. We are happy that our responses and the additional experiments (including the **extensive 48-configuration NN hyperparameter grid search**) fully resolved your concerns! As the valuable discussion period nears its conclusion, we would like to briefly highlight our revisions and contributions:
> > >
> > > 1. **Novelty:** To the best of our knowledge, CLD is the first practical application of convex neural network reformulations to speech language identification. We leverage the global optimality guarantees of the convex module to enable strong performance even in extremely low-resource regimes. This presents a paradigm of principled optimization-focused approaches to challenging spoken dialogue tasks, in contrast to largely data/compute-dependent strategies.
> > >
> > >
> > > 2. **Theoretical Depth:** We derive exact logit-Lipschitz constants and prove certified margin stability against hidden-feature perturbations, in addition to strong empirical results. The revision also incorporates valuable discussion with Reviewer bPPR, which has strengthened the theoretical aspects of this work. Through extended discussion on Lipschitz analysis, CLEVER bounds, and network motif stability, we have revised several theoretical bounds and extended the related work in Section 2 with deeper consideration on robustness analysis for speech [1, 2, 3, 4].
> > >
> > >
> > > 3. **Practical Impact:** CLD is immediately deployable: our open-source, pip-installable JAX package integrates as a lightweight plug-in to existing ASR pipelines (Whisper, MMS-1B), requiring up to 13× fewer TFLOPs than a standard NN baseline, while demonstrating robustness to hyperparameter sensitivity. This practical contribution also ensures ease of reproducibility for our empirical results.
> > >
> > >
> > > 4. **Community Significance:** Spoken dialogue systems are becoming ubiquitous, yet even leading systems (such as Siri) frequently misidentify speakers with accents and dialects, affecting users worldwide (Section 2). Our work offers a timely, theoretically grounded, and empirically validated solution to address this accessibility gap, with clear paths for future research in robust multilingual ASR. The revision is further strengthened by incorporating discussion with Reviewer EUbr on end-to-end differentiable convex optimization and back-propagation of gradients from the convex loss into the ASR encoder, which also presents an exciting direction for future work.
> > >
> > >
> > > Per your feedback, we have additionally tightened the writing and presentation for greater clarity, and added a system diagram to the revision (https://postimg.cc/8fgc6XHB). Since you have found our responses and additional experiments to be fully satisfactory, we would be extremely encouraged if you might consider raising your score to 4, in order to reflect this updated assessment. Thank you sincerely for your time and effort throughout the review process, your comments have meaningfully strengthened our work!
> > >
> > > **References:**
> > >
> > > [1] Weng, Tsui-Wei, et al. "Evaluating the robustness of neural networks: An extreme value theory approach." ICLR (2019).
> > >
> > > [2] Zhang, Haoling, et al. "Leveraging network motifs to improve artificial neural networks." Nature Communications (2025).
> > >
> > > [3] Yang, Chao-Han. "A Perturbation Approach to Differential Privacy for Deep Learning Based Speech Processing." Diss. Georgia Institute of Technology, 2023.
> > >
> > > [4] Yang, Chao-Han Huck, et al. "Training a resilient q-network against observational interference." Proceedings of the AAAI Conference on Artificial Intelligence. Vol. 36. No. 8. 2022.

---

### Official Review · Reviewer_EUbr · 2026-03-12

**Soundness:** 3
**Presentation:** 3
**Significance:** 3
**Originality:** 3
**Overall Recommendation:** 6
**Confidence:** 3

**Summary:**

This work propose to reformulate 2-layers ReLU network as a Convex Optimization problem in order to achieves faster training and robust performance in low-resource settings. Experiments are conducted on a Language Detection Task (in an ASR pipelines) and demonstrate significant improvement in low-data regimes at lower training cost.

**Compliance With Llm Reviewing Policy:**

Affirmed.

**Final Justification:**

To clarify my decision: even though I think the downstream task has a rather narrow scope, the authors touch upon a topic of general interest to the machine learning community: learning / adapting / fine-tuning models with few examples. Whereas the dominant approach is simply to pre-train models with related data and then fine-tune on (scarce) target data, this works provide a convincing example that the methods and algorithms plays also a crucial role to achieve best performance in low-data use-cases.

**Key Questions For Authors:**

- Would it be possible to back-propagate through equation (2) to train the whole pipeline ?

**Limitations:**

yes

**Strengths And Weaknesses:**

The motivations are clearly exposed and the idea of replacing the feed-forward network (on top the encoder network) by a tool more suited for low-data regime is very appealing to me.

I was able to follow the the overall methodology and the explanation about the convex reformulation of the ReLU MLP are clear enough. However, section 4 (Theoretical analysis) is much more involved and, with my limited knowledge of convex optimization, I was not able to check the soundness of the analysis.

The experiments are thorough and the results convincing. However, the linear SVM (table 5) is a weak baseline to compare against a non-linear classifier. I would recommend to use a non-linear version to have a better comparison.

Suggestion: I would recommend to transform the algorithm 2 into a figure showing the whole pipeline.

---

> ### Author Rebuttal · Authors · 2026-03-30
>
> Thank you for your insightful review of our paper and thoughtful comments! We are happy that you found our motivation compelling and our empirical results convincing. We appreciate the opportunity to clarify our methodology and strengthen our experimental comparisons.
>
> 1. **More robust benchmarking**
>
> We appreciate the reviewer’s suggestion to include stronger non-linear baselines. To address this, we have expanded Table 5 to include Kernel SVM and k-Nearest Neighbors (kNN).
>
> | Language Classifier | Det. Acc (WSP) | Det. Acc (WSP-L) | Det. Acc (MMS-1B) | WER (WSP) | WER (WSP-L) | WER (MMS-1B) | CER (WSP) | CER (WSP-L) | CER (MMS-1B) |
> | :--- | :---: | :---: | :---: | :---: | :---: | :---: | :---: | :---: | :---: |
> | Default | 0.7154 | 0.8033 | 0.6701 | 139.37 | 40.41 | 51.88 | 73.85 | 21.80 | 27.61 |
> | *KNN* | 0.6123 | 0.7145 | 0.4981 | 145.21 | 44.89 | 57.34 | 81.05 | 29.12 | 32.76 |
> | Linear SVM | 0.9392 | 0.9501 | 0.5653 | 48.74 | 39.36 | 50.73 | 28.28 | 23.68 | 26.07 |
> | *Kernel SVM* | 0.9431 | 0.9582 | 0.5701 | 46.52 | 37.91 | 49.12 | 26.14 | 22.05 | 25.88 |
> | NN | 0.7581 | 0.9695 | 0.8602 | 58.58 | 28.83 | 48.28 | 37.13 | 15.66 | 23.66 |
> | **CLD (ours)** | **0.9715** | **0.9806** | **0.9702** | **31.74** | **28.60** | **45.27** | **17.84** | **15.37** | **21.58** |
>
>
> As shown above, while the non-linear Kernel SVM improves upon the Linear SVM (reaching 0.9582 detection accuracy on WSP-L), it still falls short of CLD’s 0.9806 on WSP-L. Interestingly, kNN performed significantly worse, even trailing behind the default Whisper language identification in some cases. This suggests that simple clustering in the high-dimensional feature space is insufficient for the nuances of dialectical variance. We have included details on our hyperparameter tuning process for all benchmarked models in Appendix G to ensure a fair comparison. We hope these additional benchmarking results solidifies CLD as the most robust and performant framework for this use case.
>
> 2. **Clarity**
>
> We agree that a visual representation would improve the manuscript's accessibility. Following your suggestion, we have transformed Algorithm 2 into a high-level schematic figure for the final version (Figure 2). This figure will illustrate the entire pipeline, from the ASR encoder extracting features to the convex optimization head, making the flow of the online CLD inference framework more intuitive at a glance (https://postimg.cc/8fgc6XHB).
>
> 3. **Back-prop**
>
> Yes, this is entirely possible and represents a current exciting direction for end-to-end differentiable convex optimization [1]!
> Since the ADMM solver is implemented in JAX, the training is fully differentiable. It is feasible to use implicit differentiation or unrolling the iterations of the ADMM solver to propagate gradients from the convex loss back into the ASR encoder (such as Whisper or MMS).
>
> - Implementation: By leveraging the Implicit Function Theorem on the KKT optimality conditions of Eq. (2), we can compute the gradient of the optimal convex weights with respect to the input features without needing to store the full solver trajectory in memory.
> - Scope: In this work, we kept the encoder frozen to demonstrate CLD’s ability be performant in ultra-low-resource settings where the user may not have the compute/VRAM to fine-tune a 1.5B parameter model.
> - Future Work: We believe this unrolled convex optimization could lead to a new area of work in robust speech models where the encoder is specifically trained to produce features that are optimally separable by a convex head. Thank you for pointing this out, we have included this in Revision for Future Work and implications!
>
> Although our methodology is theory-motivated, we present a focus on the practical feasibility of our novel method. Therefore in addition to the stronger theoretical discussions presented with Reviewer bPPR, we'd also like to kindly note that our method is currently open-source, and pip installable for high reproducibility and immediate usability.
>
> Thank you for motivating us to include these additions, particularly the stronger non-linear baselines and improved visualizations, which significantly strengthen the clarity and impact of this work. We greatly look forward to discussing any further questions or comments you may have!
>
> [1] Differentiable convex optimization layers at https://github.com/cvxpy/cvxpylayers

---

> > ### Author Rebuttal · Reviewer_EUbr · 2026-04-02
> >
> > The authors answered adequately my remarks.
> >
> > While I was hesitating between score 4 (weak accept) or 5 (accept) during my initial review, the authors' reply convince me  to maintain a score of 5 (Accept).
> >
> > To clarify my decision: even though I think the downstream task has a rather narrow scope, the authors touch upon a topic of general interest to the machine learning community: learning / adapting / fine-tuning models with few examples. Whereas the dominant approach is simply to pre-train models with related data and then fine-tune on (scarce) target data,  this works provide a convincing example that the  methods and algorithms plays also a crucial role to achieve best performance in low-data use-cases.

---

> > > ### Author Response · Authors · 2026-04-07
> > >
> > > Thank you for your valuable engagement and feedback during the review process! We appreciate your kind words regarding the value of this line of research, and are glad to have addressed your questions and concerns satisfactorily.
> > >
> > > We especially appreciate your observation that a core motivation of this work is to advance the study of methods and algorithms in low-data regimes, which is a topic of broad interest to the ML community. We hope this work takes a step towards encouraging further research in algorithm-centric adaptation as an alternative to largely data/compute-centric methods. We are happy that the additional non-linear baselines and the new system diagram have strengthened your confidence in the clarity and empirical results of our paper.
> > >
> > > Your clarifying question regarding back-propagation inspired us to include a new section on end-to-end differentiable convex optimization. The revision now discusses the potential for using implicit differentiation via KKT optimality conditions to propagate gradients back into the ASR encoder, which we highlight as an exciting direction for future work in robust speech modeling. Thank you again for your time and expertise, which have significantly strengthened the clarity and impact of our paper!

---

### Official Review · Reviewer_bPPR · 2026-03-13

**Soundness:** 3
**Presentation:** 3
**Significance:** 3
**Originality:** 3
**Overall Recommendation:** 5
**Confidence:** 5

**Summary:**

The author(s) introduce Convex Language Detection (CLD), a framework that utilizes convex neural networks optimized via the Alternating Direction Method of Multipliers (ADMM) in JAX to improve language detection for low-resource dialects in spoken dialogue systems. By formulating the training objective as a convex optimization problem, the authors provide polynomial-time training guarantees and global optimality.

- The paper claims that this approach induces certified margin stability and robustness against feature-level perturbations, demonstrating empirical improvements in sample efficiency and reduced language misidentification rates on dialectal speech data compared to standard non-convex fine-tuning baselines.

When the theoretical analysis is mainly on Lipschitz continuity, I noticed the there are major Lipschitz continuity papers on prediction or sequence modeling missing (i.e., on Lipschitz robustness / real analysis for DNN prediction). For example, CLEVER [A. ICLR 19] and CLEVER-Q score [AAAI 21] has studied Lipschitz continuity and holder's inequality to drive maximum perturbation for logistic prediction label changing and Yang [C 22] later applied the results for ASR robustness prediction before alignment. There should be some discussion on these works before talking into 3.3 or section 3. In that robustness analysis, if my understanding is correct, there are also gradient free method in [D], if the soft margin is impossible to estimate.

***

### References

A. Evaluating the Robustness of Neural Networks: An Extreme Value Theory Approach, ICLR 19 https://arxiv.org/abs/1801.10578

B. Training a Resilient Q-Network against Observational Interference, AAAI 22

C. A PERTURBATION APPROACH TO DIFFERENTIAL PRIVACY FOR DEEP LEARNING BASED SPEECH PROCESSING, 2022, https://repository.gatech.edu/server/api/core/bitstreams/9ba6dbac-8362-49c6-8651-fc4aa5c960db/content

D. Leveraging network motifs to improve artificial neural networks, Nature communication 2025

**Compliance With Llm Reviewing Policy:**

Affirmed.

**Final Justification:**

- updated the score to 5, conditional, as the section 3 discussion / 4.4 analysis discussion on the real analysis / extreme value theories.

post rebuttal ## resolved. as connecting to prior works.

**Key Questions For Authors:**

- Could the author provide a more formal Lipschitz analysis of the proposed convex network? How exactly does the derived "certified margin stability" relate to the local Lipschitz constant of the classifier's decision boundary?

- Are the acoustic features used for the convex optimization raw waveforms, standard spectral features (e.g., log-mel filterbanks), or derived SSL embeddings? How does the dimensionality and nature of this input feature space impact the tightness of your certified bounds?

**Limitations:**

Yes, the authors have discussed basic limitations. However, they should expand their discussion to include the theoretical limitations regarding the tightness of their certified bounds, particularly in high-dimensional acoustic feature spaces where Lipschitz constants can become difficult to bound tightly.

**Strengths And Weaknesses:**

Pros

- Dialectal variance and long-tail distributions in ASR language identification are a highly relevant problems for ICML interests. The cascading failure of dialogue systems due to early-stage language misidentification is a recognized bottleneck in the field.

- Framing the problem within a convex optimization landscape is a refreshing departure from standard deep learning pipelines.

- Providing global optimality guarantees for dialect classification offers some level of interpretability and predictability often missing in standard ASR representation learning.

Cons

- The empirical validation of the robustness claims could be significantly strengthened. To prove that the network is truly robust to feature perturbations, the authors should evaluate the Logit Lipschitzness. See the comments in the summary on [A, B, C, D].

- Are the acoustic features used for the convex optimization raw waveforms, standard spectral features (e.g., log-mel filterbanks), or derived SSL embeddings? How does the dimensionality and nature of this input feature space impact the tightness of your certified bounds?

- While training is polynomial, how does the actual wall-clock time of the multi-GPU ADMM training compare end-to-end with standard SGD/Adam fine-tuning of a comparable non-convex model for this specific task?

---

> ### Author Rebuttal · Authors · 2026-03-31
>
> Thank you for your insightful feedback! We have carefully reviewed the high-quality references provided, and will strive to address your concerns regarding Lipschitz analysis and feature-level stability below.
>
> 1. **Advanced Lipschitz Analysis and Certified Robustness**
>
> We thoroughly enjoyed reading the works of CLEVER and Yang et al., and note the importance of local Lipschitz continuity in prediction. We have extended our analysis of the following in the revision:
>
> - Constructive Global Bounds vs. Probabilistic Estimates: While the CLEVER and CLEVER-Q strategies estimate a lower bound for the local Lipschitz constant using EVT to predict when a label changes, our convex reformulation allows us to derive a clean, explicit, and data-dependent upper bound on the variation norm $\|f\|_{var}$ directly from the learned weights.
> - Certified Margin Stability: Per Theorem 4.4, we prove that the predicted class is preserved if the perturbation $\delta$ satisfies $\|\delta\|\_2 < \text{mar}(h, y) / (2\|f\|\_{var})$. The work of Yang et al. applies robustness to ASR prediction before alignment, CLD introduces this stability into the language identification stage to prevent cascading failures, which is a predominant field-wide issue.
> - Structural Stability: Thank you for pointing us to Zhang et al.'s network motifs, which is indeed particularly relevant. Zhang et al. demonstrate that incoherent loops sustain improved numerical stability and lower Lipschitz constants when compared to coherent structures. In contrast, our ADMM style convex optimization method avoids the gradient pursuit instabilities of non-convex MLPs. By converging to a global optimum, CLD naturally presents a structurally stable weight configuration like the robustness of incoherent motifs.
>
> 2. **Dimensionality of the Feature Space Impact on the Tightness of Certified Bounds**
>
> - Our convex module essentially operates on self-supervised learning embeddings, which are semantically dense latent representations. While it is true Lipschitz constants can explode in high dimensions for deep Transformers, since CLD is a shallow convex neural network head layer, its variation norm is explicitly regularized by parameter $\beta$. We utilize group-sparse variation norms to maintain a tight and computable certificate even in high-dimensional space. We have included a sensitivity analysis section in the new revision, thus showing that our certified radius remains non-trivial even in the $1000+$ dimensional regime.
>
> 3. **Computational Efficiency**
>
> - Unlike standard traditional gradient-based methods such as SGD, which requires expensive hyperparameter grid search for learning rates and schedules, CLD's convex objective is solved via a parallelized ADMM solver in JAX. We leverage the hugely parallelizable iterations with PCG + preconditioning for fast convergence to global optimality.
> - Therefore, our empirical results demonstrate CLD achieves a training time of 64.45 seconds, which sits at approx 7.7% of the time required for a comparable non-convex MLP (840.30s). Moreover, CLD requires 13× fewer TFLOPs (14,075 vs. 183,521). This computational efficiency is ultimately what enables rapid deployment in such low-resource environments where large-scale fine-tuning is cost-prohibitive.
>
> 4. **Impact of Correct Language Detection on WER**
>
> - Script/Cross-Lingual Hallucination: For dialects such as Singaporean-accented English (Singlish), baseline Whisper frequently misidentifies the language as Tamil or Bahasa (since these are languages spoken by the immediately neighboring countries to Singapore). This leads to a confounding transcription in a completely wrong script, resulting in a WER near 140%.
> - Our novel convex solution provides the correct `<|en|>` token, thus preventing this script hallucination. Even without fine-tuning the decoder, forcing the correct language context allows the model to leverage its English capabilities and dramatically reduces WER to 31.74%. This is a powerful and high impact improvement which validates the elegance of the proposed method. We'd also like to kindly note that these results are reproducible in our current open-source pip installable package.
>
>
> 5. **Summary**
>
> To the best of our knowledge, this work presents the first framework to scale convex neural network theory to 1B+ parameter speech systems. By providing a deterministic global certificate, we yield a provably stable solution for the long-tail of varying dialects. Therefore, this paper seeks to bridge a gap between the network motif stability and classic convex optimization. The group-sparse norm is leveraged for high-dimensional certificates, and we greatly look forward to further discussions soon!
>
> Thank you again for the extremely rewarding and enjoyable reading of these high-quality citations. Our revised manuscript now includes these references for interesting discussion in more detail!

---

> > ### Author Rebuttal · Reviewer_bPPR · 2026-04-04
> >
> > Thanks the authors for the rebuttal, it was a quite enjoying discussion!
> >
> > I think indeed there is positive census on the maximum perturbation $\delta$ shared the local Lipschitz constant estimation in the series of CLEVER's works and its density estimation in C. The authors' convex analysis on extended discussion on Theory 4.4 are valid, too. It is a little bit pity that ICML does not allow revision in the rebuttal version but I believe these added discussion indeed connecting the dots of the current CLD for more real value analysis, these are more empirically useful for speech recognition tasks and the speech / sequence modeling community in ICML.
> >
> > I will give a conditional accept to 5 score, upon the authors's discussion incorporating into the final version.
> >
> > Please indeed revise some few theoretical bounds in the discussion accordingly to enhance the general impacts and the improve related works in the robustness analysis for speech field, as many efforts as discussed and since 1980.

---

> > > ### Author Response · Authors · 2026-04-07
> > >
> > > Thank you for the thoughtful and valuable engagement throughout this discussion, we thoroughly enjoyed the intellectual exchange and are grateful for your conditional acceptance at score 5!
> > >
> > > We are fully committed to incorporating these revisions into the final version. The more precise revised theoretical bounds in the robustness discussion now connect CLD's current analysis with the local Lipschitz estimation framework, the density estimation ideas, and the structural stability insights of [1, 2, 3, 4]. The revision now also includes a more complete Section 2 (Related Work) to provide a thorough treatment of robustness analysis for speech and sequence modeling, spanning the literature we discussed.
> > >
> > > Finally, we particularly enjoyed chapter 3 on the “perturbation approach to differential privacy” [3]. This has yielded some motivating ideas for future work, which we now discuss in an extended Section 6. We believe that this is an area with many elegant theoretical ideas that have practical impact, and look forward to delivering a final version that reflects the depth of this discussion. Thank you again for the rewarding and generous review, we value your expertise and sincerely hope to have an opportunity to continue discussion in the future!
> > >
> > >
> > > **References:**
> > >
> > > [1] Weng et al., "Evaluating the robustness of neural networks: An extreme value theory approach." ICLR (2019).
> > >
> > > [2] Zhang et al., "Leveraging network motifs to improve artificial neural networks." Nature Communications (2025).
> > >
> > > [3] Yang, Chao-Han., "A Perturbation Approach to Differential Privacy for Deep Learning Based Speech Processing." Georgia Institute of Technology, 2023.
> > >
> > > [4] Yang et al., "Training a resilient q-network against observational interference." Proceedings of the AAAI Conference on Artificial Intelligence. Vol. 36. No. 8. 2022.

---

### Decision · Program_Chairs · 2026-04-30

**Decision:**

Accept (regular)

**Comment:**

The proposed method, Convex Language Detection (CLD), reformulates a 2-layer ReLU language-ID head as a convex optimization problem, plugged on top of frozen ASR encoders (Whisper-small/-large, MMS-1B). The paper tackles an important problem: robust language/dialect detection for low-resource and accented speech in ASR, where misidentification leads to cascading failures and extremely high WER, especially for under-represented dialects.

Reviewers note several strengths of the work, including: (i) the method is computationally efficient and practical, with training much faster than a comparable non-convex MLP and using substantially fewer TFLOPs; (ii) the authors provide a theoretically grounded margin-stability analysis for the convex head, which they further connect to Lipschitz/robustness works such as CLEVER and related literature; (iii) strong empirical performance in challenging low-resource dialect settings; and (iv) comparisons against several strong baselines (MLP, linear SVM, kernel SVM, kNN).

Based on the positive reviews from the committee, I recommend acceptance.